# Quantum Chaos and Quantum Randomness—Paradigms of Entropy Production on the Smallest Scales

**DOI:** 10.3390/e21030286

**Published:** 2019-03-15

**Authors:** Thomas Dittrich

**Affiliations:** Departamento de Física, Universidad Nacional de Colombia, Bogotá 111321, Colombia; tdittrich@unal.edu.co; Tel.: +57-1-3165000 (ext. 10276)

**Keywords:** quantum chaos, measurement, randomness, information, decoherence, dissipation, spin, Bernoulli map, kicked rotor, standard map

## Abstract

Quantum chaos is presented as a paradigm of information processing by dynamical systems at the bottom of the range of phase-space scales. Starting with a brief review of classical chaos as entropy flow from micro- to macro-scales, I argue that quantum chaos came as an indispensable rectification, removing inconsistencies related to entropy in classical chaos: bottom-up information currents require an inexhaustible entropy production and a diverging information density in phase-space, reminiscent of Gibbs’ paradox in statistical mechanics. It is shown how a mere discretization of the state space of classical models already entails phenomena similar to hallmarks of quantum chaos and how the unitary time evolution in a closed system directly implies the “quantum death” of classical chaos. As complementary evidence, I discuss quantum chaos under continuous measurement. Here, the two-way exchange of information with a macroscopic apparatus opens an inexhaustible source of entropy and lifts the limitations implied by unitary quantum dynamics in closed systems. The infiltration of fresh entropy restores permanent chaotic dynamics in observed quantum systems. Could other instances of stochasticity in quantum mechanics be interpreted in a similar guise? Where observed quantum systems generate randomness, could it result from an exchange of entropy with the macroscopic meter? This possibility is explored, presenting a model for spin measurement in a unitary setting and some preliminary analytical results based on it.

## 1. Introduction

With the advent of the first publications proposing the concept of deterministic chaos and substantiating it with a novel tool, computer simulations, more was achieved than just a major progress in fields such as weather and turbulence [1]. They suggested a radically new view of stochastic phenomena in physics. Instead of subsuming them under a gross global category such as“chance” or “randomness”, the concept of chaos offered a profound analysis on the basis of deterministic evolution equations, thus indicating an identifiable source of stochasticity in macroscopic phenomena. A seminal insight, to be expounded in Section 2, that arose as a spin-off of the study of deterministic chaos was that the entropy produced by chaotic systems emerges by amplifying structures, initially contained in the smallest scales, to macroscopic visibility [2].

Inspired and intrigued by this idea, researchers such as Giulio Casati and Boris Chirikov saw its potential as a promising approach also towards the microscopic foundations of statistical mechanics, thus accepting the challenge to extend chaos to quantum mechanics. In the same spirit as those pioneering works on deterministic chaos, they applied standard quantization to Hamiltonian models of classical chaos and solved the corresponding Schrödinger equation numerically [3], again utilizing the powerful computing equipment available at that time. What they obtained was a complete failure on first sight. Yet, it paved the way towards a deeper understanding not only of classical chaos, but also of the principles of quantum mechanics, concerning in particular the way information is processed on atomic scales: in closed quantum systems, the entropy production characteristic of classical chaos ceases after a finite time and gives way to a behavior that is not only deterministic, but even repetitive, at least in a statistical sense; hence, it does not generate novelty any longer. The “quantum death of classical chaos” will be illustrated in Section 3.1.

The present article recalls this development, drawing attention to a third decisive aspect that is able to reconcile that striking discrepancy found between quantum and classical dynamics in closed chaotic systems. The answer that comes immediately to mind, how to bridge the gap between quantum and classical physics, is semiclassical approximations. They involve hybrid descriptions, based on a combination of quantum and classical concepts, which arise in the asymptotic regime of “ℏ→0”, referring to a relative Planck’s constant in units of characteristic classical action scales [4,5], or equivalently in a the limit of short wavelengths. Also in the case of quantum chaos, they provide valuable insight into the fingerprints classical chaotic behavior leaves in quantum systems. However, it turns out that particularly in this context, the limit ℏ→0 is not enough [6]. A more fundamental cause contributing to the discrepancy between classical and quantum chaos lies in the isolation of the employed models against their environment [7,8,9,10,11,12,13,14]. It excludes an aspect of classicality that is essential for the phenomena we observe on the macroscopic level: no quantum system is perfectly isolated, or else, we could not even know of its existence.

The role of an interaction with a macroscopic environment first came into sight in other areas where quantum mechanics appears incompatible with basic classical phenomena, such as in particular dissipation [15,16,17]. Here, even classically, irreversible behavior can only be reconciled with time-reversal invariant microscopic equations of motion if a coupling to a reservoir with a macroscopic number of degrees of freedom (or a quasi-continuous spectrum) is assumed. Quantum mechanically, this coupling not only explains an irreversible loss of energy, it leads to a second consequence, at least as fundamental as dissipation: a loss of information, which becomes manifest as decoherence [18,19].

In the context of quantum dissipation, decoherence could appear as secondary to the energy loss, yet it is the central issue in another context where quantum behavior resisted a satisfactory interpretation for a long time: quantum measurement. The “collapse of the wave packet” remained an open problem even within the framework of unitary quantum mechanics, till it could be traced back as well to the presence of a macroscopic environment, incorporated in the measurement apparatus [20,21,22,23,24,25]. As such, the collapse is not an annoying side effect, but plainly indispensable, to make sure that the measurement leaves a lasting record in the apparatus, thus becoming a fact in the sense of classical physics. Since there is no dissipation involved in this case, quantum measurement became a paradigm of decoherence induced by interaction and entanglement with an environment.

The same idea, that decoherence and the increase in entropy accompanying it is a constituent aspect of classicality, proves fruitful in the context of quantum chaos, as well [7,8,11,12,13,14]. It complements semiclassical approximations in the sense of a short-wavelength limit, in that it lifts the “splendid isolation”, which inhibits a sustained increase of entropy in closed quantum systems. Section 3.2 elucidates how the coupling to an environment restores the entropy production, constituent for deterministic chaos, at least partially in classically chaotic quantum systems. Combining decoherence with dissipation, other important facets of quantum chaos come into focus: it opens the possibility to study quantum effects also in phenomena related to dissipative chaos, notably strange attractors, which, as fractals, are incompatible with uncertainty.

The insight guiding this article is that in the context of quantum chaos, the interaction with an environment has a double-sided effect: it induces decoherence, as a loss of information, e.g., on phases of the central quantum system, but also returns entropy from the environment to the chaotic system [13,26], which then fuels its macroscopic entropy production. If indeed there is a two-way traffic, an interchange of entropy between the system and environment, this principle, applied in turn to quantum measurement, has a tantalizing consequence: it suggests that besides decoherence, besides the collapse of the wave packet, also the randomness apparent in the outcomes of quantum measurements could be traced back to the environment and could be interpreted as a manifestation of entropy interchanged with the macroscopic apparatus as a result of their entanglement during the measurement. This hypothesis is illustrated in Section 4 for the emblematic case of spin measurement. While Section 2 and Section 3 largely have the character of reviews, complementing the work of various authors with some original material, Section 4 is a perspective: it presents a project in progress at the time of writing this report.

## 2. Classical Chaos and Information Flows between Micro- and Macro-Scales

### 2.1. Overview

The relationship between dynamics and information flows has been pointed out by mathematical physicists, such as Andrey Kolmogorov, and much before deterministic chaos was (re)discovered in applied science, as is evident for example in the notion of Kolmogorov–Sinai entropy [27]. It measures the information production by a system with at least one positive Lyapunov exponent and represents a central result of research on dynamical disorder in microscopic systems, relevant primarily for statistical mechanics. For models of macroscopic chaos, typically including dissipation, an interpretation as a phenomenon that has to do with a directed information flow between scales came only much later. A seminal work in that direction is the 1980 article by Robert Shaw [2], where, in a detailed discussion in information theoretic terms, he contrasts the bottom-up information flow related to chaos with the top-down flow underlying dissipation.

Shaw argued that the contraction of phase-space area in a dissipative system results in an increasing loss of information on its initial state, if its current state is observed with a given resolution. Conversely, later states can be determined to higher and higher accuracy from measurements of the initial state. Chaotic systems show the opposite tendency: phase-space expansion, as consequence of exponentially diverging trajectories, allows one to retrodict the initial from the present state with increasing precision, while forecasting the final state requires more and more precise measurements of the initial state as their separation in time increases.

Chaotic systems therefore produce entropy, at a rate given by their Lyapunov exponents, as is also reflected in the spreading of any initial distribution of finite width. The divergence of trajectories also indicates the origin of this information: the chaotic flow amplifies the details of the initial distribution with an exponentially-increasing magnification factor. If the state of the system is observed with constant resolution, so that the total information on the present state is bounded, the gain of information on small details is accompanied by a loss of information on the largest scale, which impedes inverting the dynamics: chaotic systems are *globally* irreversible, while the irreversibility of dissipative systems is a result of their loosing *local* information into ever smaller scales.

We achieve a more complete picture already by going to Hamiltonian systems. Their phase-space flow is symplectic; it conserves phase-space area or volume, so that every expansion in some direction of phase space must be compensated by contraction in another direction. In terms of information flows, this means that an information current from small to large scales (bottom-up), corresponding to chaotic phase-space expansion [2], will be accompanied by an opposite current of the same magnitude, returning information to small scales (top-down) [2]. In the framework of Hamiltonian dynamics, however, the top-down current is not related to dissipation; it is not irreversible, but to the contrary, complements chaotic expansion in such a way that all in all, information is conserved, and the time evolution remains reversible.

A direct consequence of volume conservation by Hamiltonian flows is that Hamiltonian dynamics also *conserves entropy*; see Appendix A. As is true for the underlying conservation of volume, this invariance proves to be even more general than energy conservation and applies, e.g., also to systems with a time-dependent external force where the total energy is *not* conserved. It indicates how to integrate dissipative systems in this more comprehensive frame: dissipation and other irreversible macroscopic phenomena can be described within a Hamiltonian setting by going to models that include microscopic degrees of freedom, typically as heat baths comprising an infinite number of freedoms, on an equal footing in the equations of motion. In this way, entropy conservation applies to the entire system.

The conservation of the total entropy in systems comprising two or more degrees of freedom or subsystems cannot be reduced, however, to a global sum rule implying a simple exchange of information through currents among subsystems. The reason is that in the presence of correlations, there exists a positive amount of mutual information that prevents subdividing the total information content uniquely into contributions associated with subsystems. This notwithstanding, if the partition is not too complex, as is the case for a central system coupled to a heat bath, it is still possible to keep track of internal information flows between these two sectors. For the particular instance of dissipative chaos, a picture emerges that comprises three components:a “vertical” current from large to small scales in certain dimensions within the central system, representing the entropy loss that accompanies the dissipative loss of energy,an opposite vertical current, from small to large scales, induced by the chaotic dynamics in other dimensions of the central system,a “horizontal” exchange of information between the central system and the heat bath, including a redistribution of entropy within the reservoir, induced by its internal dynamics.

On balance, more entropy must be dumped by dissipation into the heat bath than is lifted by chaos into the central system, thus maintaining consistency with the Second Law. In phenomenological terms, this tendency is reflected in the overall contraction of a dissipative chaotic system onto a strange attractor. After transients have faded out, the chaotic dynamics then develops on a sub-manifold of reduced dimension of the phase space of the central system, the attractor. For the global information flow, it is clear that in a macroscopic chaotic system, the entropy that surfaces at large scales by chaotic phase-space expansion has partially been injected into the small scales from microscopic degrees of freedom of the environment.

Processes converting macroscopic structure into microscopic entropy, such as dissipation, are the generic case. This report, though, is dedicated to the exceptional cases, notably chaotic systems, which turn microscopic noise into macroscopic randomness. The final section is intended to show that processes even belong to this category where this is less evident, in particular quantum measurements.

### 2.2. Example 1: Bernoulli Map and Baker Map

Arguably, the simplest known model for classical deterministic chaos is the Bernoulli map [28,29], a mapping of the unit interval onto itself that deviates from linearity only by a single discontinuity,
(1)$$x\mapsto x^{\prime }=2x\;\left(\mathrm{mod}\;1\right)=\left\{\begin{array}{cc} 2x & 0\leq x<0.5,\\ 2x-1 & 0.5\leq x<1, \end{array}\right.$$
and can be interpreted as a mathematical model of a popular card-shuffling technique (Figure 1). The way it generates information by lifting it from scales too small to be resolved to macroscopic visibility becomes immediately apparent if the argument *x* is represented as a binary sequence, $x=\sum_{n=1}^{\infty }a_{n}2^{-n}$, $a_{n}\in \left\{0,1\right\}$, so that the map operates as:(2)$$x^{\prime }=2\left(\sum_{n=1}^{\infty }a_{n}2^{-n}\right)\left(\mathrm{mod}\;1\right)=\sum_{n=1}^{\infty }a_{n}2^{-n+1}\left(\mathrm{mod}\;1\right)=\sum_{n=1}^{\infty }a_{n+1}2^{-n},$$
that is, the image $x^{\prime }$ has the binary expansion:(3)$$x^{\prime }=\sum_{n=1}^{\infty }a_{n}^{\prime }2^{-n},\;\mathrm{with}\;a_{n}^{\prime }=a_{n+1}.$$

The action of the map consists of shifting the sequence of binary coefficients rigidly by one position to the left (the “Bernoulli shift”) and discarding the most significant digit $a_{1}$. In terms of information, this operation creates exactly one bit per time step, entering from the smallest resolvable scales, and at the same time, loses one bit at the largest scale (Figure 2a), which renders the map non-invertible. The spreading by a factor of two per time step can be interpreted as resulting from a continuous expansion Δx(t)=Δx(0)exp(λt), with a constant Lyapunov exponent λ=ln(2). The Bernoulli shift thus exemplifies the entropy production by chaotic systems given by the general expression:(4)$$\frac{dI}{dt}=cH_{\mathrm{KS}},\;H_{\mathrm{KS}}=\sum_{i,\;\lambda_{i}>0}\lambda_{i},$$
where *c* is a constant fixing the units of information (e.g., $c=\log_{2}\left(e\right)$ for bits and $c=k_{B}$, the Boltzmann constant, for thermodynamic entropy) and $H_{\mathrm{KS}}$ denotes the *Kolmogorov–Sinai entropy*, the sum of all *positive* Lyapunov exponents [30].

By adding another dimension, the Bernoulli map is readily complemented so as to become compatible with symplectic geometry. As the action of the map on the second coordinate, say *p*, has to compensate for the expansion by a factor of two in *x*, this suggests modeling it as a map of the unit square onto itself, contracting *p* by the same factor,
(5)$$\left(\begin{array}{c} x\\ p \end{array}\right)\mapsto \left(\begin{array}{c} x^{\prime }\\ p^{\prime } \end{array}\right),\;\left(\begin{array}{c} x^{\prime }\\ p^{\prime } \end{array}\right)=\left(\begin{array}{c} 2x\;(\mathrm{mod}\;1)\\ \frac{1}{2}\left(p+\mathrm{int}(2x)\right) \end{array}\right),$$
known as the baker map [27,29]. Geometrically, it can be interpreted as a combination of stretching (by the expanding action of the Bernoulli map) and folding (corresponding to the discontinuity of the Bernoulli map) (Figure 3). Being volume conserving, the baker map *is* invertible. The inverse map reads
(6)$$\left(\begin{array}{c} x^{\prime }\\ p^{\prime } \end{array}\right)\mapsto \left(\begin{array}{c} x\\ p \end{array}\right),\;\left(\begin{array}{c} x\\ p \end{array}\right)=\left(\begin{array}{c} \frac{1}{2}\left(x^{\prime }+\mathrm{int}\left(2p^{\prime }\right)\right)\\ 2p\;(\mathrm{mod}\;1) \end{array}\right).$$
It interchanges the operations on *x* and *p* of the forward baker map.

The information flows underlying the baker map are revealed by encoding also *p* as a binary sequence, $p=\sum_{n=1}^{\infty }b_{n}2^{-n}$. The action of the map again translates to a rigid shift,
(7)$$p^{\prime }=\sum_{n=1}^{\infty }b_{n}^{\prime }2^{-n},\;\mathrm{with}\;b_{n}^{\prime }:=\left\{\begin{array}{cc} a_{1} & n=1,\\ b_{n-1} & n\geq 2. \end{array}\right.$$
It now moves the sequence by one step *to the right*, that is from large to small scales. The most significant digit $b_{1}^{\prime }$, which is not contained in the original sequence for *p*, is transferred from the binary code for *x*; it recovers the coefficient $a_{1}$ that is discarded due to the expansion in *x*. This “paternoster mechanism” reflects the invertibility of the map. The upward information current in *x* is turned around to become a downward current in *p* (Figure 2b). A full circle cannot be closed, however, as long as the “depth” from where and to which the information current reaches remains unrestricted by any finite resolution, indicated in Figure 2, as is manifest in the infinite upper limit of the sums in Equations (Equation 2), (Equation 3) and (Equation 7).

Generalizing the baker map so as to incorporate dissipation is straightforward [28,29]: just insert a step that contracts phase space towards the origin in the momentum direction, for example preceding the stretching and folding operations of Equation (Equation 5),
(8)$$\left(\begin{array}{c} x\\ p \end{array}\right)\mapsto \left(\begin{array}{c} x^{\prime }\\ p^{\prime } \end{array}\right)=\left(\begin{array}{c} x\\ ap \end{array}\right),\;\left(\begin{array}{c} x^{\prime }\\ p^{\prime } \end{array}\right)\mapsto \left(\begin{array}{c} x^{\prime \prime }\\ p^{\prime \prime } \end{array}\right)=\left(\begin{array}{c} 2x\;(\mathrm{mod}\;1)\\ \frac{1}{2}\left(p+\mathrm{int}(2x)\right) \end{array}\right).$$
A contraction by a factor *a*, 0<a≤1, models a dissipative reduction of the momentum by the same factor. Figure 4 illustrates for the first three steps how the generalized baker map operates, starting from a homogeneous distribution over the unit square. For each step, the volume per strip reduces by a/2, while the number of strips doubles, so that the overall volume reduction is given by *a*. Asymptotically, a strange attractor emerges (rightmost panel in Figure 4) with a fractal dimension, calculated as box-counting dimension [30],
(9)$$D_{0}=\frac{\log(\mathrm{volume}\;\mathrm{contraction})}{\log(\mathrm{scale}\;\mathrm{factor})}=\frac{\ln(1/2)}{\ln(a/2)}=\frac{\ln(2)}{\ln(2)+\ln(1/a)}.$$
For example, for a=0.5, as in Figure 4, a dimension $D_{0}=0.5$ results for the vertical cross-section of the strange attractor, hence D=1.5 for the entire two-dimensional manifold.

This model of dissipative chaos is simple enough to allow for a complete balance of all information currents involved. Adopting the same binary coding as in Equation (Equation 7), a single dissipative step of the mapping, with a=0.5, Equation (Equation 8), has the effect:(10)$$p^{\prime }=\frac{p}{2}=\frac{1}{2}\sum_{n=1}^{\infty }b_{n}^{\prime }2^{-n}=\sum_{n=1}^{\infty }b_{n}^{\prime }2^{-n-1}.$$
That is, if *p* is represented as $0.\;b_{1}\;b_{2}\;b_{3}\;b_{4}\;\ldots$, $p^{\prime }$ as $0.\;b_{1}^{\prime }\;b_{2}^{\prime }\;b_{3}^{\prime }\;b_{4}^{\prime }\;\ldots$, the new binary coefficients are given by a rigid shift by one unit to the right, but with the leftmost digit replaced by zero,
(11)$$b_{n}^{\prime }=\left\{\begin{array}{cc} 0 & n=1,\\ b_{n-1} & n\geq 2. \end{array}\right.$$

Combined with the original baker map (Equation 7), this additional step fits in one digit zero each between every two binary digits transferred from position to momentum (Figure 4). In terms of information currents, this means that only half of the information lifted up by chaotic expansion in *x* returns to small scales by the compensating contraction in *p*; the other half is diverted by dissipation (Figure 5). This particularly simple picture owes itself of course to the special choice a=0.5. Still, for other values of *a*, different from 1/2 or an integer power thereof, the situation will be qualitatively the same. The fact that the dissipative information loss occurs here at the largest scales, along with the volume conserving chaotic contraction in *p*, not at the smallest as would be expected on physical grounds, is an artifact of the utterly simplified model.

### 2.3. Example 2: Kicked Rotor and Standard Map

A model that comes much closer to a physical interpretation than the Bernoulli and baker maps is the kicked rotor [27,29,31]. It can be motivated as an example, reduced to a minimum of details, of a circle map [27,29], a discrete dynamical system conceived of to describe the phase-space flow in Hamiltonian systems close to integrability. The kicked rotor, the version in continuous time of this model, can even be defined by a Hamiltonian, but allowing for a time-dependent external force,
(12)$$H\left(p,\theta ,t\right)=\frac{p^{2}}{2}+V\left(\theta \right)\sum_{n=-\infty }^{\infty }\delta \left(t-n\right),\;V\left(\theta \right)=K\cos\left(\theta \right).$$
It can be interpreted as a plane rotor with angle θ and angular momentum *p* and with unit inertia, driven by impulses that depend on the angle as a nonlinear function, a pendulum potential, and on time as a periodic chain of delta kicks of strength *K* with period one.

Reducing the continuous-time Hamiltonian (Equation 12) to a corresponding discrete-time version in the form of a map is not a unique operation, but depends, for example, on the way stroboscopic time sections are inserted relative to the kicks. If they follow immediately after each delta kick, $t_{n}=\lim_{\epsilon ↘0^{+}}\left(n+\epsilon \right)$, n∈Z, the map from $t_{n}$–$t_{n+1}$ reads:(13)$$\left(\begin{array}{c} p\\ \theta \end{array}\right)\mapsto \left(\begin{array}{c} p^{\prime }\\ \theta^{\prime } \end{array}\right),\;\left(\begin{array}{c} p^{\prime }\\ \theta^{\prime } \end{array}\right)=\left(\begin{array}{c} p+K\sin(\theta^{\prime })\\ \theta +p \end{array}\right).$$
It is often referred to as the standard or Chirikov map [27,29,31].

The dynamical scenario of this model [27,32] is by far richer than that of the Bernoulli and baker maps and constitutes a prototypical example of the Kolmogorov–Arnol’d–Moser (KAM) theorem [27]. The parameter *K* controls the deviation of the system from integrability. While for K=0, the kicked rotor is integrable, equivalent to an unperturbed circle map, increasing *K* leads to a complex sequence of mixed dynamics, with regular and chaotic phase-space regions interweaving each other in an intricate fractal structure. For large values of *K*, roughly given by K≳1, almost all regular structures in phase-space disappear, and the dynamics becomes purely chaotic. For the cylindrical phase-space of the kicked rotor, (p,θ)∈R⊗[0,2π], this means that the angle approaches a homogeneous distribution over the circle, while the angular momentum spreads diffusively over the cylinder, a case of deterministic diffusion, here induced by the randomizing action of the kicks.

For finite values of *K*, the spreading of the angular momentum does not yet follow a simple diffusion law, owing to small regular islands in phase space [33]. Asymptotically for K→∞, however, the angular momentum spreads diffusively,
(14)$$\langle {\left(p_{n}-\langle p\rangle \right)}^{2}\rangle =D\left(K\right)n$$
with a diffusion constant [27]:(15)$$D\left(K\right)=K^{2}/2$$
This regime is of particular interest in the present context, as it allows for a simple estimate of the entropy production. In the kicked rotor, information currents cannot be separated as neatly as in the baker map into a macro-micro flow in one coordinate and a micro-macro flow in the other. The complex fractal phase-space structures imply that these currents are organized differently in each point in phase space. Nevertheless, some global features, relevant for the total entropy balance, can be extracted without going into such detail.

Define a probability density in phase-space carrying the full information on the state of the system,
(16)$$\rho :\;R⊗\left[0,2\pi \right]\to R^{+},\;R⊗\left[0,2\pi \right]\;∋\left(p,\theta \right)\mapsto \rho \left(p,\theta \right)\in R^{+},\;\int_{-\infty }^{\infty }dp\int_{0}^{2\pi }d\theta \;\rho \left(p,\theta \right)=1.$$
This density evolves deterministically according to Liouville’s theorem [27,34]:(17)$$\frac{d}{dt}\rho \left(p,\theta ,t\right)=\left\{\rho (p,\theta ,t),H(p,\theta ,t)\right\}+\frac{\partial }{\partial t}\rho \left(p,\theta ,t\right),$$
involving the Poisson bracket with the Hamiltonian (Equation 12).

The kicked rotor does not conserve energy, but with its symplectic phase-space dynamics, it does conserve entropy (cf. Appendix A) if the full density distribution ρ(p,θ,t) is considered. Extracting a positive entropy production from the expanding phase-space directions alone, as with the Kolmogorov– Sinai entropy, Equation (Equation 4), will not yield significant results in a system that mixes intricately chaotic with regular regions in its phase-space. In order to obtain an overall entropy production anyway, some coarse graining is required. In the case of the kicked rotor, it offers itself to integrate ρ(p,θ,t) over θ, since the angular distribution rapidly approaches homogeneity, concealing microscopic information in fine details, while the diffusive spreading in *p* contains the most relevant large-scale structure. A time-dependent probability density for the angular momentum alone is defined projecting by the full distribution along θ,
(18)$$\rho_{p}\left(p,t\right):=\int_{0}^{2\pi }d\theta \;\rho \left(p,\theta ,t\right),\;\int_{-\infty }^{\infty }dp\;\rho_{p}\left(p,t\right)=1.$$
Its time evolution is no longer given by Equation (Equation 86), but follows a Fokker–Planck equation,
(19)$$\frac{\partial }{\partial t}\rho_{p}\left(p,t\right)=D\left(K\right)\frac{\partial^{2}}{\partial p^{2}}\rho_{p}\left(p,t\right).$$
For a localized initial condition, $\rho \left(p,0\right)=\delta \left(p-p_{0}\right)$, Equation (Equation 19), it is solved for t>0 by a Gaussian with a width that increases linearly with time:(20)$$\rho_{p}\left(p,t\right)=\frac{1}{\sqrt{2\pi }\sigma \left(t\right)}\exp\left(-\frac{{\left(p-p_{0}\right)}^{2}}{2{\left(\sigma (t)\right)}^{2}}\right),\;\sigma \left(t\right)=D\left(K\right)t.$$
Define the total information content of the density $\rho_{p}\left(p,t\right)$ as:(21)$$I\left(t\right)=-c\int_{-\infty }^{\infty }dp\;\rho_{p}\left(p,t\right)\ln\left(d_{p}\rho_{p}\left(p,t\right)\right),$$
where $d_{p}$ denotes the resolution of angular momentum measurements. The diffusive spreading given by Equation (Equation 20) corresponds to a total entropy growing as:(22)$$I\left(t\right)=\frac{c}{2}\left[\ln\left(\frac{2\pi D(K)t}{d_{p}^{2}}\right)+1\right],$$
hence to an entropy production rate dI/dt=c/2t. This positive rate decays with time, but only algebraically, that is, without a definite time scale.

The angular-momentum diffusion (Equation 14), manifest in the entropy production (Equation 22), also referred to as *deterministic diffusion* [27], is an irreversible process, yet based on a deterministic reversible evolution law. It can be reconciled with entropy conservation in Hamiltonian dynamics (Appendix A) only by assuming a simultaneous contraction in another phase-space direction that compensates for the diffusive expansion. In the case of the kicked rotor, it occurs in the angle variable θ, which stores the information lost in *p* in fine details of the density distribution, similar to the opposed information currents in the baker map (Figure 2). Indeed, this fine structure has to be erased to derive the diffusion law (Equation 14), typically by projecting along θ and neglecting autocorrelations in this variable [27].

Even if dissipation is not the central issue here, including it to illustrate a few relevant aspects in the present context is in fact straightforward. On the level of the discrete-time map, Equation (Equation 13), a linear reduction of the angular momentum leads to the dissipative standard map or Zaslavsky map [35,36],
(23)$$\left(\begin{array}{c} p\\ \theta \end{array}\right)\mapsto \left(\begin{array}{c} p^{\prime }\\ \theta^{\prime } \end{array}\right),\;\left(\begin{array}{c} p^{\prime }\\ \theta^{\prime } \end{array}\right)=\left(\begin{array}{c} e^{-\lambda }p+K\sin\left(\theta^{\prime }\right)\\ \theta +e^{-\lambda }p \end{array}\right).$$
The factor exp(−λ) results from integrating the equations of motion:(24)$$\dot{p}=-\lambda p+K\sin\left(\theta \right)\sum_{n=-\infty }^{\infty }\delta \left(t-n\right),\;\dot{\theta }=p.$$
The Fokker–Planck Equation (Equation 19) has to be complemented accordingly by a drift term ∼$\partial \rho_{p}\left(p,t\right)/\partial p$,
(25)$$\frac{\partial }{\partial t}\rho_{p}\left(p,t\right)=\left(1-\lambda \right)\frac{\partial }{\partial p}\rho_{p}\left(p,t\right)+\frac{\partial }{\partial p}\left(D\left(K\right)+{\left((1-\lambda )p\right)}^{2}\right)\frac{\partial }{\partial p}\rho_{p}\left(p,t\right).$$
In the chaotic regime K≳1 of the conservative standard map, the dissipative map (Equation 23) approaches a stationary state characterized by a strange attractor; see, e.g., [35,36].

### 2.4. Anticipating Quantum Chaos: Classical Chaos on Discrete Spaces

Classical chaos can be understood as the manifestation of information currents that lift microscopic details to macroscopic visibility [2]. Do they draw from an inexhaustible information supply on ever smaller scales? The question bears the existence of an upper bound of the information density in phase space or other physically relevant state spaces, or equivalently, on a fundamental limit of distinguishability, an issue raised, e.g., by Gibbs’ paradox [37]. Down to which difference between their states will two physical systems remain distinct? The question has already been answered implicitly above by keeping the number of binary digits in Equations (Equation 2), (Equation 3) and (Equation 7) indefinite, in agreement with the general attitude of classical mechanics not to introduce any absolute limit of distinguishability.

A similar situation arises if chaotic maps are simulated on digital machines with finite precision and/or finite memory capacity [38,39,40,41]. In order to assess the consequences of discretizing the state space of a chaotic system, impose a finite resolution in Equations (Equation 2), (Equation 3) and (Equation 7), say $d_{x}=1/J$, $J=2^{N}$ with N∈N, so that the sums over binary digits only run up to *N*. This step is motivated, for example, by returning to the card-shuffling technique quoted as the inspiration for the Bernoulli map (Figure 1). A finite number of cards, say *J*, in the card deck, corresponding to a discretization of the coordinate *x* into steps of size $d_{x}>0$, will substantially alter the dynamics of the model.

More precisely, specify the discrete coordinate as:(26)$$x_{j}=\frac{j-1}{J},\;j=1,\;2,\;3,\;\ldots ,\;J,\;J=2^{N},\;N\in N,$$
with a binary code $x=\sum_{n=1}^{N}a_{n}2^{-n}$. A density distribution over the discrete space $\left(x_{1},x_{2},\ldots ,x_{J}\right)$ can now be written as a *J*-dimensional vector:(27)$$\rho =\left(\rho_{1},\rho_{2},\rho_{3},\ldots ,\rho_{J}\right),\;\rho_{j}\in R^{+},\;\sum_{j=1}^{N}\rho_{j}=1,$$
so that the Bernoulli map takes the form of a (J×J)-permutation matrix $B_{J}$, $\rho \mapsto \rho^{\prime }=B_{J}\rho$. These matrices reproduce the graph of the Bernoulli map (Figure 1), but discretized on a (J×J) square grid. Moreover, they incorporate a deterministic version of the step of interlacing two partial card decks in the shuffling procedure, in an alternating sequence resembling a zipper. For example, for J=8, N=3, the matrix reads:(28)$$B_{8}=\left(\begin{array}{cccccccc} 1 & 0 & 0 & 0 & 0 & 0 & 0 & 0\\ 0 & 0 & 0 & 0 & 1 & 0 & 0 & 0\\ 0 & 1 & 0 & 0 & 0 & 0 & 0 & 0\\ 0 & 0 & 0 & 0 & 0 & 1 & 0 & 0\\ 0 & 0 & 1 & 0 & 0 & 0 & 0 & 0\\ 0 & 0 & 0 & 0 & 0 & 0 & 1 & 0\\ 0 & 0 & 0 & 1 & 0 & 0 & 0 & 0\\ 0 & 0 & 0 & 0 & 0 & 0 & 0 & 1 \end{array}\right).$$
The two sets of entries =1 along slanted vertical lines represent the two branches of the graph in Figure 1, as shown in Figure 6b.

A deterministic dynamics on a discrete state space comprising a finite number of states must repeat after a finite number *M* of steps, no larger than the total number of states. In the case of the Bernoulli map, the recursion time is easy to calculate: in binary digits, the position discretized to $2^{N}$ bins is specified by a sequence of *N* binary coefficients $a_{n}$. The Bernoulli shift moves this entire sequence in M=N=lb(J) steps, which is the period of the map. Exactly how the reshuffling of the cards leads to the full recovery or the initial state after *M* steps is illustrated in Figure 7. That is, the shuffling undoes itself after *M* repetitions!

The time when the discrete map crosses over from chaotic to periodic behavior grows as the logarithm of the system size. This anticipates the *Ehrenfest time scale* of quantum chaos [13,42], transferred to the classical level: Take a quantum system with an initially close-to-classical dynamics expanding in some phase-space direction as exp(λt), with a rate given by the Lyapunov exponent λ. In a rough estimate, one expects deviations from classical chaotic spreading to occur when an initial structure of the size of a Planck cell *ℏ* reaches the size of a characteristic classical action *A*, that is at a time $t^{*}=\lambda^{-1}\ln\left(A/\hbar \right)$ (for a precise, conceptually more consistent definition of this time scale, see [13]). To apply this reasoning to the discrete Bernoulli map, replace the total number of Planck cells A/ℏ by the number of states *J* and set λ=ln(2), the number of bits generated per time step, resulting in M=lb(J).

A similar, but even more striking situation occurs for the baker map, discretized in the same fashion. While the *x*-component is identical to the discrete Bernoulli map, the *p*-component is construed as the inverse of the *x*-component; cf. Equation (Equation 6). Defining a matrix of probabilities on the discrete (J×J) square grid that replaces the continuous phase-space of the baker map,
(29)$$\rho :\;\left\{1,\;\ldots ,\;J\right\}⊗\left\{1,\;\ldots ,\;J\right\}\to R^{+},\;\left(n,m\right)\mapsto \rho_{n,m},\;\sum_{n,m=1}^{J}\rho_{n,m}=1,$$
the discrete map takes the form of a similarity transformation,
(30)$$\rho \mapsto \rho^{\prime }=B_{J}^{-1}\rho B_{J}^{t}=B_{J}^{t}\rho B_{J}^{t}.$$
The inverse matrix $B_{J}^{-1}$ is readily obtained as the transpose of $B_{J}$. For example, for N=3, it reads:(31)$$B_{8}^{-1}=B_{8}^{t}=\left(\begin{array}{cccccccc} 1 & 0 & 0 & 0 & 0 & 0 & 0 & 0\\ 0 & 0 & 1 & 0 & 0 & 0 & 0 & 0\\ 0 & 0 & 0 & 0 & 1 & 0 & 0 & 0\\ 0 & 0 & 0 & 0 & 0 & 0 & 1 & 0\\ 0 & 1 & 0 & 0 & 0 & 0 & 0 & 0\\ 0 & 0 & 0 & 1 & 0 & 0 & 0 & 0\\ 0 & 0 & 0 & 0 & 0 & 1 & 0 & 0\\ 0 & 0 & 0 & 0 & 0 & 0 & 0 & 1 \end{array}\right).$$
As for the forward discrete map, it resembles the corresponding continuous graph (Figure 6a), with entries one now aligned along two slanted horizontal lines (Figure 6b).

Both the upward shift of binary digits of the *x*-component and the downward shift of binary digits encoding *p* now become periodic with period M=N, as for the discrete baker map. The two opposing information currents thus close to a circle, resembling a paternoster lift with a lower turning point at the least significant and an upper turning point at the most significant digit (Figure 8). It is to be emphasized that the map (Equation 5), being deterministic and reversible, conserves entropy, which implies a zero entropy production rate. The fact that the discrete baker map is no longer chaotic but periodic therefore does not depend on the vanishing entropy production, but reflects the finite total information content of its discrete state space.

The fate of deterministic classical chaos in systems comprising only a finite number of discrete states (of a “granular phase space”) has been studied in various systems [38,39,40,41], with the same general conclusion that chaotic entropy production gives way to periodic behavior with a period determined by the size of the discrete state space, that is by the finite precision underlying its discretization. To a certain extent, this classical phenomenon anticipates the effects of quantization on chaotic dynamics, but it provides at most a caricature of quantum chaos. It takes only a single, if crucial, tenet of quantum mechanics into account, the fundamental bound uncertainty imposes on the storage density of information in phase space, leaving all other principles of quantum mechanics aside. Yet, it shares a central feature with quantum chaos, the repetitive character it attains in closed systems, and it suggests how to interpret this phenomenon in terms of information flows.

## 3. Quantum Death and Incoherent Resurrection of Classical Chaos

While the “poor man’s quantization” discussed in the previous section indicates qualitatively what to expect if chaos is discretized, reconstructing classically chaotic systems systematically in the framework of quantum mechanics allows for a much more profound analysis of how these systems process information (for comprehensive bibliographies on quantum chaos in general, readers are kindly asked to consult monographs such as [4,43,44,45]). Quantum mechanics directs our view more specifically to the aspect of closure of dynamical systems. Chaotic systems provide a particularly sensitive probe, more so than systems with a regular classical mechanics, of the effects of a complete blocking of external sources of entropy, since they react even to a weak coupling to the environment by a radical change of their dynamical behavior.

### 3.1. Quantum Chaos in Closed Systems

In this section, prototypical examples of the quantum suppression of chaos will be contrasted with open systems where classical behavior reemerges at least partially. A straightforward strategy to study the effect first principles of quantum mechanics have on chaotic dynamics is quantizing models of classical chaos. This requires these models, however, to be furnished with a minimum of mathematical structure, required for a quantum mechanical description. In essence, systems with a volume-conserving flow, generated by a classical Hamiltonian on an even-dimensional state space can be readily quantized. In the following, the principal consequences of quantizing chaos will be exemplified applying this strategy to the baker map and the kicked rotor.

In quantum mechanics, the finiteness of the space accessible to a system has very distinctive consequences: The spectrum of eigenenergies becomes discrete, a feature directly related to quasi-periodic evolution in the time domain, incompatible with chaotic behavior. The discreteness of the spectrum therefore is a sufficient condition for the quantum suppression of chaos. Still, the ability of a quantum system to imitate classical chaos at least for a finite time implies more specific properties of the discrete spectrum. They become manifest in the statistics of two-point and higher spectral correlations and can be studied in terms of different symmetry classes of random matrices and their eigenvalue spectra. Spectral statistics is not of direct relevance in the present context of entropy production; readers interested in the subject are referred to monographs such as [32,45].

#### 3.1.1. The Quantized Baker Map

The baker map introduced in Section 2.2 is an ideal model to consider quantum chaos in a minimalist setting. It already comprises a coordinate together with its canonically conjugate momentum and can be quantized in an elegant fashion [46,47,48]. Starting from the operators $\hat{x}$ and $\hat{p}$, $\hat{p}=-i\hbar d/dx$ in the position representation, with commutator $[\hat{x},\hat{p}]=i\hbar$, their eigenspaces are constructed as:(32)$$\hat{x}|x\rangle =x|x\rangle ,\;\hat{p}|p\rangle =p|p\rangle ,\;\langle x|p\rangle =\frac{e^{ipx/\hbar }}{\sqrt{2\pi \hbar }}.$$
The finite classical phase-space $\left[0,1\right]⊗[0,1]\;\subset \;R^{2}$ of the baker map can be implemented with this pair of quantum operators by assuming periodicity, say with period one, both in *x* and in *p*. Periodicity in *x* entails quantization of *p* and vice versa, so that together, a Hilbert space of finite dimension *J* results, and the pair of eigenspaces (Equation 32) is replaced by:(33)$$\hat{x}|j\rangle =\frac{j}{J}|j\rangle ,\;\hat{p}|l\rangle =\hbar l|l\rangle ,\;j,\;l=0,\;\ldots ,\;J-1,\;\langle j|l\rangle =\frac{1}{\sqrt{J}}\;e^{2\pi i\;jl/J}={\left(F_{J}\right)}_{j,l},$$
that is the transformation between the two spaces coincides with the discrete Fourier transform, given by the (J×J)-matrix $F_{J}$.

This construction suggests a straightforward quantization of the baker map. If we phrase the classical map as the sequence of actions:expand the unit square [0,1]⊗[0,1] by a factor of two in *x*,divide the expanded *x*-interval into two equal sections, [0,1] and [1,2],shift the right one of the two rectangles (Figure 3), (x,p)∈[1,2]⊗[0,1], by one to the left in *x* and by one up in *p*, [1,2]⊗[0,1]↦[0,1]⊗[1,2],contract by two in *p*,
this translates to the following operations on the Hilbert space defined in Equation (Equation 33), assuming the Hilbert-space dimension *J* to be even,in the *x*-representation, divide the vector of coefficients $\left(a_{0},\;\ldots ,\;a_{J-1}\right)$, $|x\rangle =\sum_{j=0}^{J-1}a_{j}|j\rangle$, into two halves, $\left(a_{0},\;\ldots ,\;a_{J/2-1}\right)$ and $\left(a_{J/2},\;\ldots ,\;a_{J-1}\right)$,transform both partial vectors separately to the *p*-representation, applying a $\left(\frac{J}{2}\times \frac{J}{2}\right)$-Fourier transform to each of them,stack the Fourier transformed right half column vector on top of the Fourier transformed left half, so as to represent the upper half of the spectrum of spatial frequencies,transform the combined state vector from the *J*-dimensional *p*-representation back to the *x* representation, applying an inverse (J×J)-Fourier transform.

All in all, this sequence of operations combines to a single unitary transformation matrix. In the position representation, it reads:(34)$$B_{J}^{\left(x\right)}=F_{J}^{-1}\left(\begin{array}{cc} F_{J/2} & 0\\ 0 & F_{J/2} \end{array}\right).$$

Like this, it already represents a compact quantum version of the Baker map [46,47]. It still bears one weakness, however: the origin (j,l)=(0,0) of the quantum position-momentum index space, coinciding with the classical origin (x,p)=(0,0) of phase-space, creates an asymmetry, as the diagonally opposite corner $\frac{1}{J}\left(j,l\right)=\frac{1}{J}\left(J-1,J-1\right)=\left(1-\frac{1}{J},1-\frac{1}{J}\right)$ does *not* coincide with (x,p)=(1,1). In particular, it breaks the symmetry x→1−x, p→1−p of the classical map. This symmetry can be recovered on the quantum side by a slight modification [48] of the discrete Fourier transform mediating between position and momentum representation, a shift by $\frac{1}{2}$ of the two discrete grids. It replaces $F_{J}$ by:(35)$$\langle j|l\rangle =\frac{1}{\sqrt{J}}\exp\left(2\pi i\left(j+\frac{1}{2}\right)\left(l+\frac{1}{2}\right)\right)=:{\left(G_{J}\right)}_{j,l},$$
and likewise for $F_{J/2}$. The quantum baker map in position representation becomes accordingly:(36)$$B_{J}^{\left(x\right)}=G_{J}^{-1}\left(\begin{array}{cc} G_{J/2} & 0\\ 0 & G_{J/2} \end{array}\right).$$
In momentum representation, it reads $B_{J}^{\left(p\right)}=G_{J}B_{J}^{\left(x\right)}G_{J}^{-1}$. The matrix $B_{J}^{\left(x\right)}$ exhibits the same basic structure as its classical counterpart, the *x*-component of the discrete baker map (Equation 28), but replaces the sharp “crests” along the graph of the original mapping by smooth maxima (Figure 6c). Moreover, its entries are now complex. In momentum representation, the matrix $B_{J}^{\left(p\right)}$ correspondingly resembles the *p*-component of the discrete baker map.

While the discretized classical baker map (Equation 30) merely permutes the elements of the classical phase-space distribution, the quantum baker map rotates complex state vectors in a Hilbert space of finite dimension *J*. We cannot expect periodic exact revivals as for the classical discretization. Instead, the quantum map is quasi-periodic, owing to phases $\epsilon_{n}$ of its unimodular eigenvalues $e^{i\epsilon_{n}}$, which in general are not commensurate. With a spectrum comprising a finite number of discrete frequencies, the quantum baker map therefore exhibits an irregular sequence of approximate revivals. They can be visualized by recording the return probability,
(37)$$P_{\mathrm{ret}}\left(n\right)={\left|\mathrm{Tr}[{\hat{U}}^{n}]\right|}^{2}$$
with the one-step unitary evolution operator $\langle j|\hat{U}|j^{\prime }\rangle ={\left(B_{J}^{\left(x\right)}\right)}_{j,j^{\prime }}$. Figure 9a shows the return probability of the (8×8) quantum baker map for the first 500 time steps. Several near-revivals are visible; the figure also shows the unitary transformation matrix ${\left(B_{J}^{\left(x\right)}\right)}^{n}$ for n=490 where it comes close to the (8×8) unit matrix (Figure 9b). The quantum baker map therefore does not exhibit as exact and periodic recurrences as does the discretized classical map (Figure 8), but it is evident that its dynamics deviates dramatically from the exponential decay of the return probability, constituent for mixing, hence for strongly chaotic systems [27,28,29].

This example suggests concluding that the decisive condition to suppress chaos is the finiteness of the state space, exemplified by a discrete classical phase-space or a finite-dimensional Hilbert space. Could we therefore hope chaotic behavior to be more faithfully reproduced in quantum systems with an infinite-dimensional Hilbert space? The following example frustrates this expectation, but the coherence effects, which prevent chaos also here, require a more sophisticated analysis.

#### 3.1.2. The Quantum Kicked Rotor

In contrast with mathematical toy models such as the baker map, the kicked rotor allows including most of the features of a fully-fledged Hamiltonian dynamical system, also in its quantization. With the Hamiltonian (Equation 12), a unitary time-evolution operator over a single period of the driving is readily construed [3,49]. Placing, as for the classical map, time sections immediately after each kick, the time-evolution operator reads:(38)$${\hat{U}}_{\mathrm{QKR}}={\hat{U}}_{\mathrm{kick}}{\hat{U}}_{\mathrm{rot}},\;{\hat{U}}_{\mathrm{kick}}=\exp\left(-ik\cos(\hat{\theta })\right),\;{\hat{U}}_{\mathrm{rot}}=\exp\left(-i{\hat{p}}^{2}/2\hbar \right).$$
The parameter *k* relates to the classical kick strength as k=K/ℏ. Angular momentum $\hat{p}$ and angle $\hat{\theta }$ are now operators canonically conjugate to one another, with commutator $[\hat{p},\hat{\theta }]=-i\hbar$. The Hilbert space pertaining to this model is of infinite dimension, spanned for example by the eigenstates of $\hat{p}$,
(39)$$\hat{p}|l\rangle =\hbar l|l\rangle ,\;l\in Z,\;\langle \theta |l\rangle =\frac{1}{\sqrt{2\pi \hbar }}\exp\left(il\theta \right).$$

Operating on an infinite-dimensional Hilbert space, the arguments explaining a discrete spectrum of the quantum baker map, hence quasi-periodicity of its time evolution, do not carry over immediately to the kicked rotor. On the contrary, in the quantum kicked rotor with its external driving, energy is not conserved; the system should explore the entire angular-momentum space as in the classical case, and in the regime of strong kicking, one expects to see a similar unbounded growth of the kinetic energy as symptom of chaotic diffusion as in the classical standard map. It was all the more surprising for Casati et al. [3,49] that their numerical experiments proved the opposite: the linear increase of the kinetic energy ceases after a finite number of kicks and gives way to an approximately steady state, with the kinetic energy fluctuating in a quasi-periodic manner around a constant mean value (Figure 10).

It turns out that an infinite Hilbert space dimension is not sufficient to enable a chaotic time evolution. In addition, a continuous eigenenergy spectrum is required. While for the quantum baker map, this is evidently not the case, it is far from obvious how anything like a discrete spectrum could arise in a driven system such as the kicked rotor. Eigenenergies in the usual sense of a time-dependent Hamiltonian cannot be defined. However, as the driving is invariant under discrete translations of time, t→t+1, another conservation law applies: Floquet theory [50,51] guarantees the existence of *quasienergy states*, eigenstates of ${\hat{U}}_{\mathrm{QKR}}$ with unimodular eigenvalues exp(iϵ). An explanation for quasi-periodicity was found in the quasienergy spectrum and eigenstates of the system, calculated by numerical diagonalization of ${\hat{U}}_{\mathrm{QKR}}$ [52,53,54,55]. Results clearly indicate that for generic parameter values, the spectrum of quasienergies ϵ is discrete, and the *effective* Hilbert space, accessed from a localized initial condition, is always of only finite dimension. Eigenstates $|\phi (\epsilon_{n})\rangle$ are not extended in angular-momentum space, let alone periodic. On average and superposed with strong fluctuations, they are *localized*: they decay exponentially from a center $l_{c}\left(\epsilon_{n}\right)$, different for each eigenstate,
(40)$${\left|\langle l|\phi \left(\epsilon_{n}\right)\rangle \right|}^{2}∼\exp\left(-\frac{|l-l_{c}\left(\epsilon_{n}\right)|}{L}\right).$$
The scale of this decay, the *localization length*
*L*, is generic and approximately given by $L\approx {\left(K/2\pi \hbar \right)}^{2}$, hence grows linearly with the classical diffusion constant cf. Equation (Equation 15).

This unexpected phenomenon, called *dynamical localization*, resembles Anderson localization, a coherence effect known from solid-state physics [56,57]: if a crystalline substance is disturbed by sufficiently strong “frozen disorder” (impurities, lattice dislocations, etc.), its energy eigenstates are not extended, as predicted by Bloch’s theorem [58] for a spatially-periodic potential. Rather, the plane waves corresponding to Bloch states, scattered at aperiodic defects, superpose on average destructively, so that extended states compatible with the periodicity of the potential cannot build up. In the kicked rotor, the disorder required to prevent extended states, not in position, but in angular-momentum space, does not arise by any static randomness of a potential, as in an imperfect crystal lattice, nor is it a consequence of the dynamical disorder of the chaotic classical map. It comes about by a dynamical coherence effect related to the nature of the sequence of phases $\phi \left(l\right)=\hbar l^{2}\left(\mathrm{mod}\;2\pi \right)$ of the factor ${\hat{U}}_{\mathrm{rot}}=\exp\left(-i{\hat{p}}^{2}/2\hbar \right)=\exp\left(-i\hbar {\hat{l}}^{2}/2\right)$ of the Floquet operator (Equation 38). If Planck’s constant (in the present context, *ℏ* enters as a dimensionless parameter in units of the inertia of the rotor and the period of the kicks) is not commensurable with 2π, these phases, as functions of the index *l*, constitute a pseudo-random sequence. In one dimension, this disorder of number-theoretical origin is strong enough to prevent extended eigenstates. Since the rationals form a dense subset, but of measure zero, of the real axis, an irrational value of ℏ/2π is the generic case.

Even embedded in an infinite-dimensional Hilbert space, exponential localization reduces the effective Hilbert-space dimension to a finite number $D_{H}$, determined by the number of quasienergy eigenstates that overlap appreciably with a given initial state. For an initial state sharply localized in *l*, say $\langle l|\psi \left(0\right)\rangle =\delta_{l-l_{0}}$, it is given on average by $D_{H}=2L$. This explains the crossover from chaotic diffusion to localization described above: in the basis of localized eigenstates, a sharp initial state overlaps with approximately 2L quasienergy states, resulting in the same number of complex expansion coefficients. The initial “conspiration” of their phases, required to construct the initial state $\left|\psi \left(0\right)\rangle =\right|l_{0}\rangle$, then disintegrates increasingly, with the envelope of the evolving state widening diffusively until all phases of the contributing eigenstates have lost their correlation with the initial state, at a time $n^{*}\approx 2L$, in number of kicks. The evolving state has then reached an exponential envelope, similar to the shape of the eigenstates, Equation (Equation 40) (Figure 11, dashed lines), and its width fluctuates in a pseudo-random fashion, as implied by the superposition of the 2L complex coefficients involved.

The crossover time $n^{*}$ is immediately related to the discrete character of the quasienergy spectrum. The effectively 2L quasienergies are distributed approximately homogeneously around the unit circle (a consequence of level repulsion in classically non-integrable systems [32,45]) with a mean separation of Δϵ≈2π/2L. This discreteness is resolved and becomes manifest in the time evolution, according to the energy-time uncertainty relation, after $n^{*}=2\pi /\Delta \epsilon \approx 2L$ periods of the driving. This is an example of the *Heisenberg time*. For an autonomous classically-chaotic quantum system, it predicts the crossover from chaotic to quasi-periodic behavior for a time $t^{*}\approx 2\pi \hbar /\Delta E$, if the average spectral separation is ΔE. Also here, it is related to the effective Hilbert space dimension $D_{H}$: if the total spectral range available to the system, defined by global constraints, is $E_{\max}$, then $\Delta E=E_{\max}/D_{H}$, and thus, $t^{*}\approx 2\pi \hbar D_{H}/E_{\max}$.

This scenario might appear as an exceptional effect, arising by the coincidence of various special circumstances. Indeed, there exist a number of details and exceptions, omitted in the present discussion, which lead to a different dynamical behavior, such as accelerator modes in the classical model [33,60] and quantum resonances for rational values of ℏ/2π [61]. This notwithstanding, similar studies of other models have accumulated overwhelming evidence that in quantum systems evolving as a unitary dynamics, a permanent entropy production as in classical chaos is excluded.

In more abstract terms, the “quantum death of classical chaos” can be understood as the consequence of two fundamental principles: the conservation of information under unitary time evolution, cf. Appendix B, a conservation law closely analogous to information conservation under classical canonical transformations (Appendix A), and the condition that the Hilbert space reachable from the initial state by a unitary dynamics has a finite dimension, i.e., amounts to a limited information content. Once the system has explored its entire accessible Hilbert space, it cannot but return, at least approximately, to states it had already assumed previously. Like this, it applies also to the quantum kicked rotor: even with a periodic driving, its time evolution is unitary. It is energetically open, but entropically closed.

Based merely on spectral discreteness, not on correlations, the Heisenberg time does not refer specifically to the classically chaotic nature of a quantum system. However, the information theoretic interpretation applies also to the crossover time $n^{*}$. As in the classical system, a positive entropy production can only be identified by considering the reduced angular-momentum distribution, instead of the full quantum state. In the presence of localization, the dimension of the Hilbert space effectively accessible by an initial condition local in angular momentum is $D_{H}\approx 2L$. The maximum information content it could achieve is thus given by a homogeneous distribution over $D_{H}$ states, hence by $I_{\max}\approx c\ln\left(2L\right)$. Comparing this with the entropy production by chaotic diffusion, Equation (Equation 22), allows estimating the crossover time $n^{*}$ as the time till this maximum is reached. By equating:(41)$$I_{\max}=I\left(n^{*}\right)=c\left[\ln\left(\sqrt{2\pi D\left(K\right)n^{*}}/d_{p}\right)+\frac{1}{2}\right]$$
and setting $d_{p}=\hbar$, the angular-momentum quantum, and $D\left(K\right)=K^{2}/2$, (cf. Equation (Equation 15)), it is found to be:(42)$$n^{*}\approx \frac{4}{\pi e}K^{2}.$$
This estimate, based on entropy production, coincides exactly, as to the dependence on *K*, with similar estimates based on the energy-time uncertainty relation, as well as with numerical data, which give:(43)$$n^{*}\approx 2L\approx \frac{K^{2}}{2\pi^{2}\hbar^{2}},$$
and it agrees in order of magnitude even with the prefactor.

### 3.2. Breaking the Splendid Isolation: Quantum Chaos and Quantum Measurement

If the absence of permanent entropy production in closed quantum systems is interpreted as a manifestation of quantum coherence, it is natural to inquire how immune this effect is to incoherent processes. They occur in a huge variety of circumstances: in quantum systems embedded in a material environment, as in molecular and solid state physics, interacting with a radiation field, as in quantum optics, in dissipative quantum systems where decoherence accompanies an irreversible energy loss, and most notably in all instances of observation, be it by measurement in a laboratory or by leaving any kind of permanent record in the environment [62], even in the absence of a human observer.

Dissipation is the more common context where in quantum systems, decoherence is unavoidable, but it is invariably accompanied by the main effect, energy loss. Quantum measurement, by contrast, allows separating decoherence, as an exchange of entropy with the environment, from the loss of energy. It has been in the focus of quantum theory from the early pioneering years on, providing the indispensable interface with the macroscopic world. The crucial step from quantum superpositions to alternative classical facts remained an enigma for decades. The Copenhagen interpretation includes the “collapse of the wave packet” as an essential element [63], but treats it as an unquestionable postulate. The first systematic analysis of quantum measurement by von Neumann [64] already provides a quantitative description in terms of the density operator, rendering the wave packet collapse explicit as a reduction of the density matrix to its diagonal elements, but does not yet illuminate the physical nature of this step, manifestly incompatible with the Schrödinger equation. It was the contribution of Zurek and others [20,21,22,23,65] to interpret this process, in the spirit of quantum dissipation, as the consequence of the interaction with the macroscopic number of degrees of freedom the measurement apparatus (the “meter”) and its environment comprise, to be described in a microscopic model as a heat bath or reservoir. As one of the major implications of this picture, the collapse of the wave packet no longer appears as an unstructured point-like event, but as a continuous process that can be resolved in time [23].

#### 3.2.1. Modeling Continuous Measurements on the Quantum Kicked Rotor

In this subsection, basic elements of this scheme will be adopted and applied to the quantum kicked rotor in order to demonstrate how observation can thaw dynamical localization and thus restore, at least partially, an entropy production as in classical chaos. Reducing quantum measurement to the essential, a continuous observation of the kicked rotor will be assumed, which leads to a lasting record of a suitable observable [66]. Following established models of quantum measurement [20,21,22,23,26,65], these features can be incorporated in a object-meter interaction Hamiltonian [59,67,68]:(44)$$H_{\mathrm{OM}}=g\;{\hat{x}}_{M}\;{\hat{x}}_{O}\;\Theta \left(t\right),$$
where *g* controls the coupling strength and the Heaviside function Θ(t) switches the measurement on at t=0. The operator ${\hat{x}}_{M}$, acting on the Hilbert space of the meter, is the observable that indicates the measurement result (its “pointer operator” [20,21,22,23]), and ${\hat{x}}_{O}$ is the measured observable. In accord with the objective to study the impact of observation on localization in angular momentum space, we shall focus on measurements of the angular momentum $\hat{l}$. If the expectation 〈l〉 is observed as a global measure, this amounts to defining the measured operator as:(45)$${\hat{x}}_{O}=\hat{l}=\sum_{l=-\infty }^{\infty }l|l\rangle \langle l|.$$
Alternatively, a simultaneous observation of the full angular-momentum distribution P(l), so that the measurement affects homogeneously the entire angular momentum axis, requires assuming a separate meter component ${\hat{x}}_{M,l}$ for every eigenvalue of the angular momentum,
(46)$$H_{\mathrm{OM}}=g\;{\hat{x}}_{M}\cdot {\hat{x}}_{O}\;\Theta \left(t\right)=g\sum_{l=-\infty }^{\infty }{\hat{x}}_{M,l}{\hat{x}}_{S,l},\;{\hat{x}}_{S,l}=|l\rangle \langle l|.$$
Some models of quantum measurement distinguish explicitly between the meter proper, as a microscopic system interacting directly with the observed object, and a macroscopic apparatus that couples in turn to the meter [65], thus only indirectly to the object. Such a distinction is not necessary in the present context; it suffices to merge meter and environment into a single macroscopic system. Moreover, we do not conceive a detailed microscopic model of the meter as a heat bath (but see Section 4.2 and Section 4.3 below), starting instead directly from an evolution equation that takes the essential consequences of the meter’s macroscopic nature into account.

From this basic setup, assuming standard properties of the heat bath such as an immediate response (Markovianity), evolution equations for the reduced density operator of the object, ${\hat{\rho }}_{O}\left(t\right)={\mathrm{Tr}}_{M}\left(\hat{\rho }\left(t\right)\right)$, can be construed; see Appendix C. Integrated over one time step of the driven dynamics, they take the form of maps for the density operator. In the case of measurements of 〈l〉 with the pointer observable as in Equation (Equation 45), the map for the density matrix in angular-momentum representation reads:(47)$$\langle l|{\hat{\rho }}_{S,n+1}|m\rangle =\sum_{l^{\prime },m^{\prime }=-\infty }^{\infty }b_{l^{\prime }-l}\left(k\right)b_{m^{\prime }-m}^{*}\left(k\right)\exp\left(-\frac{i\hbar }{2}\left(l^{\prime 2}-m^{\prime 2}\right)-\gamma {\left(l^{\prime }-m^{\prime }\right)}^{2}\right)\langle l^{\prime }|{\hat{\rho }}_{S,n}|l^{\prime }\rangle ,$$
while for measurements of P(l), Equation (Equation 46):(48)$$\begin{array}{cc} \langle l|{\hat{\rho }}_{S,n+1}|m\rangle =\sum_{l^{\prime },m^{\prime }=-\infty }^{\infty } & b_{l^{\prime }-l}\left(k\right)b_{m^{\prime }-m}^{*}\left(k\right)\\ & \left[\exp\left(-\frac{i\hbar }{2}\left({l^{\prime }}^{2}-{m^{\prime }}^{2}\right)\right)-e^{-\gamma }\left(1-\delta_{m^{\prime }-l^{\prime }}\right)+\delta_{m^{\prime }-l^{\prime }}\right]\langle l^{\prime }|{\hat{\rho }}_{S,n}|m^{\prime }\rangle . \end{array}$$

These maps alternate the unitary time evolution of the quantum kicked rotor with incoherent steps that lead to a gradual decay of the non-diagonal elements of the density matrix. In the limit of strong effective coupling to the meter, γ≫1, corresponding to a high-accuracy measurement of the angular momentum, the density matrix is completely diagonalized anew at each time step, and the object system leaves the measurement in an incoherent superposition of angular-momentum states, as required by the principles of quantum measurement (Figure 11b and Figure 12b). For a weaker coupling, the loss of coherence per step is only partial, restricting the density matrix to a diagonal band with a Gaussian profile of width ∼$\gamma^{-1}$, if 〈l〉 is measured, or reducing its off-diagonal elements homogeneously by $e^{-\gamma }$, if the full distribution is recorded (Figure 11a and Figure 12a). In any case, decoherence in the angular momentum representation is equivalent to a diffusive spreading of the angle θ. It imitates the action of classical chaos in that it effectively destroys the autocorrelation of the angle variable.

The framework set by Equation (Equation 76) is easily extended to include dissipation [69,70,71]. An additional term, proportional to the friction constant λ,
(49)$$\begin{array}{cc} {\dot{\hat{\rho }}}_{O}= & -\frac{i}{\hbar }\left[{\hat{H}}_{O},{\hat{\rho }}_{O}\right]+\gamma \left[{\hat{x}}_{O},\left[{\hat{\rho }}_{O},{\hat{x}}_{O}\right]\right]\\ & +\frac{1}{2}g^{2}\lambda \left(\left[{\hat{x}}_{O}{\hat{\rho }}_{O},\left[{\hat{H}}_{O},{\hat{x}}_{O}\right]\right]-\left[\left[H_{O},{\hat{x}}_{O}\right],{\hat{\rho }}_{O}{\hat{x}}_{O}\right]\right), \end{array}$$
induces incoherent transitions between angular momentum eigenstates towards lower values of *l*, modeling Ohmic friction with a damping constant λ, as in the classical standard map with dissipation, Equations (Equation 23)–(Equation 25) [35,36]. In terms of a classical stochastic dynamics, to be detailed in Appendix C, it corresponds to a drift of the probability density in phase-space towards lower angular momentum.

Describing the quantum dynamics in terms of a master equation for the reduced density operator only provides a global statistical account. However, in the semiclassical regime of small angular momentum quantum *ℏ*, compared to the periodicity of the classical phase-space in the same observable *p*, it can be replaced by an approximate description as a classical Langevin equation with a noise term of quantum origin that induces diffusion in θ [59,67,68]. Including again Ohmic friction with damping constant λ, it can be cast in the form of a classical map with noise term ξ (see Appendix C),
(50)$$\left(\begin{array}{c} p_{n+1}\\ \theta_{n+1} \end{array}\right)=\left(\begin{array}{c} p_{n}+K\sin\left(\theta_{n+1}\right)\\ \theta_{n}+e^{-\lambda }p_{n}+\xi_{n} \end{array}\right).$$

#### 3.2.2. Numerical Results

Numerical experiments performed with both, the quantum map for the density matrix, Equations (Equation 47) and (Equation 48), and its semiclassical approximation, Equations (Equation 50) and (Equation 82), give a detailed picture of the effect of continuous observation on quantum chaos [59,67,68]. Figure 12 compares the time dependence of the mean kinetic energy for the quantum kicked rotor, Equation (Equation 38) (dashed lines), the same system under continuous measurement, Equation (Equation 48) (solid lines), and the stochastic classical map, Equations (Equation 82) and (Equation 83) (dotted). Above all, the data shown provide clear evidence that *incoherent processes induced by measurements destroy dynamical localization*. Even for weak coupling to the apparatus, Figure 11a and Figure 12a, classical angular momentum diffusion is recovered, albeit on a time scale $n_{c}\approx \nu^{-1}$, much larger than the cross-over time $n^{*}$, cf. Equation (Equation 43), if ν≪1/2L, and with a diffusion constant $D_{\mathrm{qm}}\approx D\left(K\right)n^{*}/n_{c}$, reduced accordingly with respect to its classical value D(K). For stronger coupling, the measurement-induced diffusion approaches the classical strength D(K). Since it randomizes the angle variable indiscriminately, erasing all fine structure in classical phase space, it ignores deviations of D(K) from the gross estimate (Equation 15), caused, e.g., by accelerator modes of the classical standard map [33,60]. In fact, measurement-induced diffusion occurs already for kick strengths $K<K_{c}$, below the classical threshold to chaotic diffusion $K_{c}\approx 1$, where in the classical map, diffusion is still blocked by regular tori extending across the full range θ∈[0,2π[. Moreover, Figure 13b, showing the angular momentum distribution after 512 time steps, demonstrates that at this stage, the typical exp(−|l|/L) shape indicating localization has given way to a Gaussian envelope, characteristic of diffusion.

Figure 13 compares the angular momentum reached after 512 time steps for the measured quantum system in the description by the master Equation (Equation 48) (dotted lines) with that obtained for the noisy map (Equation 82) (solid lines). For sufficiently strong coupling, Figure 13b, it is faithfully reproduced by the semiclassical Langevin Equation (Equation 82), as is the overall energy growth; see Figure 12b (dotted line).

The diffusion constant of the measurement-induced angular momentum diffusion also allows us to estimate directly the entropy produced by the measured quantum system: Replacing in Equation (Equation 22) the classical diffusion constant D(K) by the reduced quantum mechanical value $D_{\mathrm{qm}}$ yields:(51)$$I\left(t\right)=\frac{c}{2}\left[\ln\left(\frac{2\pi D_{\mathrm{qm}}t}{d_{p}^{2}}\right)+2\ln\left(\frac{n^{*}}{n_{c}}\right)+1\right].$$
As the production rate for diffusive spreading is independent of the diffusion constant, it is here the same as for the classical standard map, $\dot{I}\left(t\right)=c/2t$. Such a positive entropy production is not compatible with entropy conservation in closed quantum systems; Appendix B. The only possible explanation therefore refers to the measured quantum system *not* being closed, so that the entropy generated actually infiltrates from the macroscopic meter to which it is coupled. This interpretation becomes plausible also considering the fact that obviously, there must be an entropy flow from the object towards the meter, or else the measured data could not reach it: There is no reason why the information current from object to meter should not be accompanied by an opposite current, from meter to object.

We can now distinguish three phases of, in particular, the weakly measured (i.e., with small coupling to the meter) quantum kicked rotor and interpret them from the point of view of entropy flows:During the initial phase, $n≲n^{*}$, the quantum map follows closely the classical standard map, producing entropy from its own supply provided by the initial state.Once it is exhausted, at the crossover time $n^{*}$, entropy production stalls, the system localizes and its time evolution becomes quasi-periodic. The Heisenberg time $n^{*}$ therefore marks the *upper* limit in time for a behavior of a closed quantum system imitating classical dynamics.Only on a much longer time scale, defined by the decoherence time $n_{c}\gg n^{*}$, sufficient entropy can infiltrate from the environment, here the meter, to become manifest again in the dynamics of the kicked rotor as diffusive angular-momentum spreading. Getting entangled with the environment by the measurement, the kicked rotor effectively attains an infinite Hilbert-space dimension and a continuous spectrum, despite dynamical localization, which restores a behavior close to classical chaos. In short, the decoherence time is the *lower* limit in time for an open quantum system to approach a macroscopic classical dynamics.

While decoherence alone allowed substantiating the crucial rôle of the environment, inducing chaotic dynamics in quantum systems, incorporating friction gives us the opportunity to take a look also at the modifications of *dissipative* classical chaos that are required by quantization. Here, it is a static phenomenon, the fractal geometry of strange attractors, that collides with quantum mechanics: The infinite structural depth implied by self-similarity is incompatible with uncertainty. In order to “quantize strange attractors”, the master Equation (Equation 49) as well as the stochastic semiclassical approximation, Equation (Equation 50), can be solved numerically and compared with the classical dissipative standard map (Equation 23) [69,70,71]. Figure 14 depicts the stationary states approached by these maps for n≫1/λ, the time scale of contraction onto the attractor. The classical strange attractor, Figure 14a, here represented as its support in (p,θ) phase-space, roughly follows a (−sinθ)-curve. The stationary state of the full quantum master equation, depicted as the Wigner function corresponding to the stationary density operator, Figure 14c, shows a smoothed structure that eliminates the self-similarity of the classical fractal geometry. The wavy modulations visible in Panel (c) are owed to the tendency of Wigner function to exhibit fringes where it takes negative values, if the support of the positive regions is strongly curved. They are absent in the stationary state of the semiclassical noisy map, Panel (b).

## 4. Quantum Measurement and Quantum Randomness in a Unitary Setting

In the examples discussed in the preceding sections, the central issue was chaotic entropy production and its suppression by coherence effects in closed quantum systems. Measurement served as a particular case of interaction with a macroscopic environment, giving rise to a two-way exchange of information. A transfer of information on the state of the object is the essence of measurement. It does not even require a human observer, the physical environment can play the rôle of the “witness” [62]. Conversely, entropy entering the measured object from the side of the apparatus imparts a stochastic component to the proper dynamics of the object [26]. Quantum chaos is specially sensitive to this effect, as even minuscule amounts of entropy penetrating from outside become manifest in the long-time behavior. The reason for this sensitivity lies in the unbounded amplification of perturbations, a global instability chaotic systems show throughout their state space.

The present section takes up this idea to explore its consequences in the context where its relevance is far less obvious. In quantum measurement, instabilities of the measurement process itself, instead of a sensitive dependence on initial conditions of a measured chaotic system, let us expect similar effects as in the case of quantum chaos. It is not obvious, though, where in the context of measurement such instabilities should exist, of a kind even remotely comparable to chaotic dynamics. To see this, a final step has to be added to the above outline of the quantum measurement process that had not yet been taken into account in Section 3.2.

### 4.1. Quantum Randomness from Quantum Measurement

The collapse of the wave packet is not only incompatible with a unitary time evolution, it also violates the conservation of entropy (Appendix B). If the measured system is initiated in a pure state,
(52)$$|\psi_{O,\mathrm{ini}}\rangle =\sum_{\alpha }a_{\alpha }|\alpha \rangle ,$$
(assuming a discrete basis of eigenstates of the measured operator, e.g., $\hat{x}|\alpha \rangle =x_{\alpha }|\alpha \rangle$, α∈Z) a complete collapse leads to a mixed state comprising the same components,
(53)$${\hat{\rho }}_{O,\mathrm{ini}}=|\psi_{O,\mathrm{ini}}\rangle \langle \psi_{O,\mathrm{ini}}|\to {\hat{\rho }}_{O,\mathrm{clps}}=\sum_{\alpha }p_{\alpha }|\alpha \rangle \langle \alpha |,\;p_{\alpha }={\left|a_{\alpha }\right|}^{2}.$$
The increase in entropy from the pure initial state ($I_{\mathrm{ini}}=0$) is thus:(54)$$I_{\mathrm{clps}}=-c\mathrm{Tr}\left({\hat{\rho }}_{O,\mathrm{clps}}\ln\left({\hat{\rho }}_{O,\mathrm{clps}}\right)\right)=-c\sum_{\alpha }p_{\alpha }\ln\left(p_{\alpha }\right).$$
It is readily explained and can be modeled in microscopic detail as a consequence of the entanglement of the object with the macroscopic apparatus [20,21,22,23,26,65]. It means that during this phase of the measurement, both components share their entropy, so that it can no longer be uniquely partitioned into a meter part and an object part. In the reduced density operator of the object, but likewise in that for the meter, this correlation becomes manifest as information gain: $I_{\mathrm{clps}}>I_{O,\mathrm{ini}}+I_{M,\mathrm{ini}}$. The reduced density operator of the object, “collapsed” to its diagonal, $\langle \alpha |{\hat{\rho }}_{O,\mathrm{clps}}|\alpha^{\prime }\rangle =p_{\alpha }\delta_{\alpha^{\prime }-\alpha }$, is interpreted as a set of probabilities $p_{\alpha }$ for the measurement resulting in the eigenvalue $x_{\alpha }$ of the measured operator $\hat{x}$.

With this step, the measurement is not yet complete. From the Copenhagen interpretation onwards [63], all quantum measurement schemes add a crucial final transition, to the object exiting the process again in a pure state, one of the eigenstates |α〉,
(55)ρ^O,clps=∑αpα|α〉〈α|→ρ^O,fin=|α〉⋮〈α|⋮|α〉〈α|withprobabilitypα|α〉⋮〈α|⋮
returning the information content to its initial value, $I_{O,\mathrm{fin}}=I_{O,\mathrm{ini}}=0$. This step is sometimes referred to as “second collapse of the wave packet”. In contrast to the “first collapse”, though, it is usually considered to be of little interest for the discussion of fundamentals of quantum mechanics, since it appears as a mere classical random process, analogous to drawing from an urn. There is, however, also a quantum mechanical side to it. With the second collapse, the object gets disentangled from the meter again, but there is absolutely no reason why the entropy previously shared between them should be partitioned afterwards in the same way as it had been before the measurement. Information can have been interchanged among the two systems. On the side of the object, it becomes manifest as the random process behind the phrase “with probability $p_{\alpha }$”.

This applies at least to all measurements of operators with a discrete spectrum, such as, for example, the angular momentum $\hat{l}$ of the kicked rotor. It becomes particularly evident in the case of operators on finite-dimensional Hilbert spaces, notably and as the simplest possible instance, two-state systems (“qbits”), say H=span{|↓〉,|↑〉}, ${\hat{\sigma }}_{z}|\;↓\rangle =-\frac{\hbar }{2}|\;↓\rangle$, ${\hat{\sigma }}_{z}|\;↑\rangle =\frac{\hbar }{2}|\;↑\rangle$. Preparing it as a Schrödinger cat, neutral with respect to measurements of $hat\sigma_{z}$,
(56)$$|\psi_{O,\mathrm{ini}}\rangle =\frac{1}{\sqrt{2}}(|\;\;↓\rangle \pm |\;↑\rangle \rangle ),$$
the results |↓〉〈↓| and |↑〉〈↑| are expected with equal probabilities $p_{↓}=p_{↑}=0.5$. While each outcome is a pure state with definite eigenvalue, repeated measurements of an ensemble of systems in the same initial state result in a random binary sequence, distinguished as “quantum randomness” and considered unpredictable in a more fundamental sense than any classical stochastic process [72]. The von Neumann entropy, as canonical measure of the information contained in a quantum system, is not able to capture the difference between a pure state resulting from a deterministic preparation and an element of a sequence of pure states which, as an ensemble, represent a prototypical random process.

The mere existence of a set of privileged states, the eigenstates of the measured operator (forming the “pointer basis”, a term coined by Zurek [20,21,22,23]), of course does not imply any instability. To be sure, the conservation under unitary transformations of the overlap 〈ϕ|ψ〉 as a measure of distance between two states |ψ〉, |ϕ〉 ensures that there cannot be any attractors or repellers in Hilbert space [73]. This situation changes, however, as soon as the non-unitary dynamics of incoherent processes in the projective Hilbert space is concerned. In quantum measurement, in particular, the *quantum Zeno effect* [74,75] plays a pivotal rôle [21]: if a measurement is made on a state vector that is about to rotate away from a pointer-basis state it has been prepared in, for example by a previous measurement of the same observable, this subsequent measurement will project the state back to the nearest pointer basis state as indicated by Equation (Equation 55) [20,21,22,23], i.e., the state from which it just departed. The more frequently the same measurement is being repeated, the stronger will be its stabilizing effect towards the initial pointer state: it thus becomes an attractor in the projective Hilbert space of the measured object [20,21].

If there is not just a single such state but a finite or even countably infinite number of attractors, it is clear that their basins of attraction in projective Hilbert space must be separated by boundaries, manifolds along which the system is unstable. For example, for a two-state system, the projective Hilbert space is the Bloch sphere, its poles representing the pointer states, hence the attractors for measurements of the vertical spin component (Figure 15). Symmetry already implies that the boundary separating their basins of attraction, the two hemispheres, must be the equator, representing the manifold all Schrödinger-cat states as defined in Equation (Equation 56). Of course, the attraction towards the poles is strongest in their immediate neighborhood, but vanishes for states orthogonal to the pointer states, as applies to all states along the equator.

The description in terms of an evolution equation for the density operator, such as the master Equation (Equation 76), however, does not allow going beyond stating likelihoods, in this example equal probabilities for the two outcomes. Otherwise, it leaves the second collapse as a black box. A more profound analysis is possible, though, by going to a detailed microscopic account of the coupled object-meter system. Since this comprehensive system is closed as a whole, it not only permits a description in the framework of unitary time evolution. The conservation of entropy moreover opens the possibility to follow the information interchanged between the two subsystems.

### 4.2. Spin Measurement in a Unitary Setting

The setup sketched in Section 3.2.1 is a suitable starting point for a model of measurements on a two-state system. In order to include a microscopic account of the meter, it is broken down into a set of, say, harmonic oscillators with frequencies $\omega_{n}$. The measurement object now reduces to a spin-$\frac{1}{2}$ system. Modifying the object-meter coupling, Equations (Equation 45) and (Equation 46) accordingly, it now takes the form:(57)$$H_{\mathrm{OM}}=\sum_{n}g_{n}{\hat{\sigma }}_{z}\left({\hat{a}}_{n}^{†}+{\hat{a}}_{n}\right)\;\Theta \left(t\right),$$
where the measured observable is specified as ${\hat{x}}_{O}={\hat{\sigma }}_{z}$, the vertical spin component, coupled with a strength $g_{n}$ to meter operators ${\hat{x}}_{M,n}={\hat{a}}_{n}^{†}+{\hat{a}}_{n}$ (the position operators of the $n^{\mathrm{th}}$ mode of the meter, up to a factor $\sqrt{2}$). Complemented by self-energies $H_{O}=\frac{1}{2}\hbar \omega_{0}{\hat{\sigma }}_{x}$ of the object and $H_{M}=\sum_{n}\hbar \omega_{n}\left({\hat{a}}_{n}^{†}{\hat{a}}_{n}+\frac{1}{2}\right)$ of the meter, the total Hamiltonian for the measurement process is obtained as:(58)$$\begin{array}{cc} H & =H_{O}+H_{\mathrm{OM}}+H_{M}\\ & =\frac{1}{2}\hbar \omega_{0}{\hat{\sigma }}_{x}+\sum_{n}g_{n}{\hat{\sigma }}_{z}\left({\hat{a}}_{n}^{†}+{\hat{a}}_{n}\right)\;\Theta \left(t\right)+\sum_{n}\hbar \omega_{n}\left({\hat{a}}_{n}^{†}{\hat{a}}_{n}+\frac{1}{2}\right) \end{array}$$
In terms of quantum optics, for instance, it can be interpreted as describing a two-level atom interacting with a microwave cavity supporting discrete modes *n* [76].

The model is not complete without specifying the initial state of the total system. Supposing that it factorizes between object and meter [20,21,64,65],
(59)$$|\Psi_{\mathrm{ini}}\rangle =|\psi_{O,\mathrm{ini}}\rangle |\psi_{M,\mathrm{ini}}\rangle ,$$
the initial states of the two components can be defined separately. For the object, assume a state that is neutral with respect to measurements of ${\hat{\sigma }}_{z}$, as in Equation (Equation 56). The initial state of the meter should not introduce a spatial bias of position or momentum, either, so that $\langle {\hat{x}}_{M,\mathrm{ini}}\rangle =0$, $\langle {\hat{p}}_{M,\mathrm{ini}}\rangle =0$. Otherwise it can be an arbitrary coherent superposition of harmonic oscillator states.

A crucial issue concerning Hamiltonian and initial condition is their symmetry under spatial reflections z→−z with respect to the direction of the vertical spin component. The total Hamiltonian, as well as the initial state of the object should be invariant under this transformation, otherwise the measurement would be biased. This symmetry is equivalent to parity in the *z*-direction, effectuated by operators ${\hat{\Pi }}_{z,S}={\hat{\sigma }}_{x}$ for the two-state system and ${\hat{\Pi }}_{z,M}=\exp\left(i\pi \sum_{n}{\hat{a}}_{n}^{†}{\hat{a}}_{n}\right)$ for the meter [77], so that the total system must be invariant under the transformation:(60)$${\hat{\Pi }}_{z}={\hat{\Pi }}_{z,S}{\hat{\Pi }}_{z,M}={\hat{\sigma }}_{x}\exp\left(i\pi \sum_{n}{\hat{a}}_{n}^{†}{\hat{a}}_{n}\right).$$
Indeed, it is readily verified that ${\hat{\Pi }}_{z,S}^{†}{\hat{H}}_{O}{\hat{\Pi }}_{z,S}={\hat{H}}_{O}$, ${\hat{\Pi }}_{z,M}^{†}{\hat{H}}_{M}{\hat{\Pi }}_{z,M}={\hat{H}}_{M}$, and:(61)$$\begin{array}{cc} {\hat{\Pi }}_{z}^{†}{\hat{H}}_{\mathrm{OM}}{\hat{\Pi }}_{z} & ={\hat{\Pi }}_{z,S}^{†}{\hat{\sigma }}_{z}{\hat{\Pi }}_{z,S}\sum_{n}g_{n}{\hat{\Pi }}_{z,M}^{†}\left({\hat{a}}_{n}^{†}+{\hat{a}}_{n}\right){\hat{\Pi }}_{z,M}\;\Theta \left(t\right)\\ & =\left(-{\hat{\sigma }}_{z}\right)\left(-\sum_{n}g_{n}\left({\hat{a}}_{n}^{†}+{\hat{a}}_{n}\right)\right)\Theta \left(t\right)={\hat{H}}_{\mathrm{OM}}. \end{array}$$
Given this invariance, the Hilbert space of the total system decomposes into two eigen-subspaces of ${\hat{\Pi }}_{z}$,
(62)$$H=H_{+}⊗H_{-},$$
$H_{+}$ comprising symmetric, $H_{-}$ antisymmetric states under ${\hat{\Pi }}_{z}$. As the object (two-state) as well as the meter (boson) sector of the total system can be decomposed individually into an even and an odd subspace, the parity subspaces decompose further into:(63)$$\begin{array}{cc} H_{+} & =H_{S,+}⊗H_{M,+}⊕H_{S,-}⊗H_{M,-},\\ H_{-} & =H_{S,+}⊗H_{M,-}⊕H_{S,-}⊗H_{M,+}. \end{array}$$

At the same time, both possible measurement outcomes, |↓〉, as well as |↑〉, manifestly break the invariance under z→−z individually, even if on average, they are balanced. In the framework of a unitary time evolution, where the Hamiltonian, as well as the initial state of the object are symmetric, the only possible explanation is that the asymmetry is introduced by the initial state of the meter.

Reconstructing the measurement in a unitary account of the full object-meter system allows us to pursue the time evolution of the total state vector in continuous time. Yet, it is desirable, in order to compare with the standard view of quantum measurement, to record diagnostics that enable assessing the progress towards a definite classical outcome. Two aspects are of particular significance for this purpose: the approach of the spin component towards a pure state is reflected in the time dependence of the von Neumann entropy [64] of the reduced density operator
(64)$$I_{O}\left(t\right)=-c{\mathrm{Tr}}_{O}\left[{\hat{\rho }}_{O}\left(t\right)\ln\left({\hat{\rho }}_{O}\left(t\right)\right)\right],\;{\hat{\rho }}_{O}\left(t\right)={\mathrm{Tr}}_{M}\left[\hat{\rho }\left(t\right)\right],$$
and can be quantified as its purity, $P_{O}\left(t\right)={\mathrm{Tr}}_{O}\left[{\hat{\rho }}_{O}^{2}\left(t\right)\right]$. Representing ${\hat{\rho }}_{O}\left(t\right)$ as a Bloch vector $a=(a_{x},a_{y},a_{z})$, $a_{x}:=\frac{1}{2}\mathrm{Tr}\left({\hat{\rho }}_{O}{\hat{\sigma }}_{x}\right)$ etc., the purity is reflected in its length, $P_{O}\left(t\right)=\frac{1}{2}{\left(1+|a\right|}^{2})$. The asymmetry of the spin state with respect to *z*-parity can be expressed as its polarization,
(65)$$a_{z}\left(t\right)=\frac{1}{2}\left(\rho_{↑↑}\left(t\right)-\rho_{↓↓}\left(t\right)\right)=\frac{1}{2}\left(\langle ↑\;|\hat{\rho }\left(t\right)|\;↑\rangle -\langle ↓\;|\hat{\rho }\left(t\right)|\;↓\rangle \right),$$
that is as the vertical (*z*-) component of the Bloch vector.

### 4.3. Simulating Decoherence by Finite Heat Baths

An essential condition to achieve an irreversible loss of coherence in a system coupled to a macroscopic environment is that the spectrum of the environment, be it composed of harmonic oscillators, spins [78], or other suitable microscopic models, be continuous on the energy scales of the central system, or equivalently, that the number *N* of modes the environment comprises be large, N≫1. As a general rule, based on energy-time uncertainty, recurrences occur on a time scale 1/Δω if the spectrum exhibits structures on the scale Δω. However, in the present context of a unitary model for quantum measurement, it is more appropriate to stop short of the limit N→∞. Evidently, this can be achieved only if irreversibility as a hallmark of decoherence is sacrificed.

This price appears acceptable, though, as long as a faithful description of the processes of interest is required only over a correspondingly large, but finite time scale, as is the case, for example, in computational molecular physics and in quantum optics. Numerical experiments simulating decoherence with heat baths of finite Hilbert space dimension [79,80,81] provide convincing evidence that even with a surprisingly low number of bath modes, *N* of the order of 10, most relevant features of decoherence can be satisfactorily reproduced; see Figure 16. This suggests to restrict the dimension of the meter sector of the Hilbert space underlying the Hamiltonian (Equation 58) accordingly to a finite number *N*,
(66)$$H=\frac{1}{2}\hbar \omega_{0}{\hat{\sigma }}_{x}+\sum_{n=1}^{N}g_{n}{\hat{\sigma }}_{z}\left({\hat{a}}_{n}^{†}+{\hat{a}}_{n}\right)\;\Theta \left(t\right)+\sum_{n=1}^{N}\hbar \omega_{n}\left({\hat{a}}_{n}^{†}{\hat{a}}_{n}+\frac{1}{2}\right).$$
Like this, the Hamiltonian can be considered as a model of, e.g., a two-level atom in a high-*Q* microwave cavity [76]. The mode number *N* thus assumes the rôle of a central parameter of the model.

Experience with similar models comprising finite baths [79,81], suggests the following scenario:For small values N≳1, the time evolution comprises only a few, but typically incommensurate, frequencies and should appear quasi-periodic.Already for moderate numbers, say N=O(10), the unitary model will exhibit a similar behavior as has been observed for standard models of quantum optics and solid-state physics, known as “collapses and revivals” [76]. In particular, the Zeno effect implies that the object state approaches one of the pointer states and remains in its vicinity for a longer time, before it may jump to another (in the case of spin measurement, the opposite) pointer state.For N≫1, the excursions of the object state away from pointer states will in general become smaller, while the frequency of full switching episodes—spin flips in the case of spin measurements—should reduce, that is the times the object spends close to a pointer state should grow very large. In particular, as soon as the object state is sufficiently close to one of the pointer states, a behavior reminiscent of the quantum Zeno effect should emerge [21].

In fact, a similar scenario has been predicted for a model in the spirit of quantum optics, representing the object by a two-state atom and meter and environment, respectively, by two microwave cavities coupled through a waveguide [76].

Of practical interest is the opposite extreme, N=1, as it allows us to study some issues analytically that are no longer so readily accessible for higher values of *N*. The Hamiltonian:(67)$$H_{\mathrm{sb}}=\frac{1}{2}\hbar \omega_{0}{\hat{\sigma }}_{x}+g{\hat{\sigma }}_{z}\left({\hat{a}}^{†}+\hat{a}\right)\;\Theta \left(t\right)+\hbar \omega_{1}\left({\hat{a}}^{†}\hat{a}+\frac{1}{2}\right).$$
also referred to as the *spin-boson Hamiltonian* or *quantum Rabi model* [82,83], is frequently employed as the standard model for two-level atoms interacting with a bosonic field. It is often considered in a slightly simplified version: if a rotating-wave approximation is applied that excludes double excitation or de-excitation processes (generated by ${\hat{\sigma }}_{+}{\hat{a}}^{†}$ or ${\hat{\sigma }}_{-}\hat{a}$), the interaction term reduces to ${\hat{H}}_{\mathrm{OM}}=g\left({\hat{\sigma }}_{+}\hat{a}+{\hat{\sigma }}_{-}{\hat{a}}^{†}\right)$, denoting ${\hat{\sigma }}_{\pm }:=\frac{1}{2}\left({\hat{\sigma }}_{x}\mp i{\hat{\sigma }}_{y}\right)$. With this modification, the spin-boson Hamiltonian is also known as *Jaynes–Cummings model*. The emblematic feature exhibited by spin-boson systems are *Rabi oscillations*, oscillations of the two-state system between its lower and its upper level (in our present notation, the eigenstates of ${\hat{\sigma }}_{x}$) with a frequency proportional to the coupling *g*. A further simplification of Equation (Equation 67), often called *semi-classical Rabi model*, replaces the coupling to the boson mode with frequency $\omega_{1}$ by an external driving with the same frequency [84,85], $H_{\mathrm{scl}}=\frac{1}{2}\hbar \omega_{0}{\hat{\sigma }}_{x}+g{\hat{\sigma }}_{z}\cos\left(\omega_{1}t\right)$.

With the Hamiltonian (Equation 67), it is straightforward to specify parity eigen-subspaces, referred to in Equation (Equation 63). The even eigenspace comprises states of the form:(68)$$|\Psi_{++}\rangle =\frac{1}{\sqrt{2}}\left(|\;\;↓\rangle +|\;\;↑\rangle \right)\sum_{\alpha =0}^{\infty }c_{2\alpha }|2\alpha \rangle \;\mathrm{or}\;|\Psi_{--}\rangle =\frac{1}{\sqrt{2}}\left(|\;\;↓\rangle -|\;\;↑\rangle \right)\sum_{\alpha =0}^{\infty }c_{2\alpha +1}|2\alpha +1\rangle ,$$
the odd subspace is spanned by states of the form:(69)$$|\Psi_{+-}\rangle =\frac{1}{\sqrt{2}}\left(|\;\;↓\rangle +|\;\;↑\rangle \right)\sum_{\alpha =0}^{\infty }c_{2\alpha +1}|2\alpha +1\rangle \;\mathrm{or}\;|\Psi_{-+}\rangle =\frac{1}{\sqrt{2}}\left(|\;\;↓\rangle -|\;\;↑\rangle \right)\sum_{\alpha =0}^{\infty }c_{2\alpha }|2\alpha \rangle .$$

Preliminary numerical results for the quantum dynamics, generated both by the Jaynes–Cummings model [86] and by the complete spin-boson Hamiltonian [82,83], in a parameter regime relevant for the present modeling, in particular for strong coupling, are consistent with the expectations pointed out here. For the present application to quantum measurement, there is no obvious justification for a rotating-wave approximation. With the full Hamiltonian (Equation 67), the von Neumann equation for the density operator, $i\hbar \;d\hat{\rho }/dt=\left[H_{\mathrm{sb}},\hat{\rho }\right]$ is readily evaluated at t=0 (Appendix D). For an initial state as in Equation (Equation 59), which factorizes into a Schrödinger cat for the two-state system and an arbitrary superposition of boson excitations,
(70)$$|\Psi_{\pm }\left(0\right)\rangle =\frac{1}{\sqrt{2}}\left(|\;\;↓\rangle \pm |\;\;↑\rangle \right)\sum_{\alpha =0}^{\infty }c_{\alpha }|\alpha \rangle$$
This yields for the initial tendency of the polarization $a_{z}=\frac{1}{2}\langle {\hat{\sigma }}_{z}\rangle$,
(71)$$\frac{d}{dt}a_{z}\left(t\right){|}_{t=0}=\frac{1}{2}\left({\dot{\hat{\rho }}}_{↑↑}\left(0\right)-{\dot{\hat{\rho }}}_{↓↓}\left(0\right)\right)=0.$$
That is, to leading order, the state vector starts rotating around the *z*-axis of the Bloch sphere, but does not leave the equator. However, going to the second time derivative, one finds:(72)d2dt2az(t)|t=0=12ρ^¨↑↑(0)−ρ^¨↓↓(0)=±2gω0∑α=0∞α+1Re(cα+1cα*).

This result indicates that to second order in time, a state prepared as a Schrödinger cat with respect to vertical spin will exhibit polarization if the initial state of the boson fulfills a specific condition: The terms in the sum over α in Equation (Equation 72) only contribute if not all products $c_{\alpha +1}c_{\alpha }^{*}$ of two subsequent expansion coefficients vanish. It has an obvious interpretation in terms of symmetry: the boson components in the eigen-subspaces of the parity operator ${\hat{\Pi }}_{z}$, Equations (Equation 68) and (Equation 69), are characterized by encompassing exclusively even or exclusively odd components of each sector, spin and boson, of the total system. The condition $c_{\alpha +1}c_{\alpha }^{*}\neq 0$ for the boson sector therefore implies that the initial state *of the meter* must not belong to either one of the two eigen-subspaces $H_{+}$ and $H_{-}$, hence must break z→−z parity, while the initial state of the spin itself has to remain unbiased.

Combining these analytic findings with the quantum Zeno effect (Section 4.1) allows predicting that initial states, unbiased as to spin polarization, will, to leading order in time, rotate along the equator of the Bloch sphere, the attraction basin boundary between spin-up and spin-down, but to higher order move away from it in a direction depending on an asymmetric initial state of the meter, to become attracted by that pole of the Bloch sphere they are already approaching; see Figure 15.

Following a similar research program as in quantum chaos, comparing quantum dynamics to its closest classical analogue, it would be tempting to study the unitary model for spin measurement in some appropriate classical limit. A model based on a symmetric double-well potential, closely analogous in many respects to a spin measurement, can be devised that already provides relevant insights, as sketched in Appendix E. A similar model for a classical binary “random” process, a coin toss, has been analyzed in all detail in [87]. Diaconis et al. studied the basin boundaries separating initial conditions of the coin that lead to either one of the two outcomes “head” and “tail”. They show a conspicuous structure of alternating fine fringes corresponding to these final conditions. While in the case of coin tosses, the sensitive dependence on the initial state of the coin itself serves as random generator, it is the initial state of the environment that generates randomness in the double-well model.

### 4.4. Perspectives

A unitary account of quantum measurements with random outcome, as outlined in this section, is presently being worked out. Starting from the analytical framework presented here, it requires massive numerical calculations. The quantum model with finite mode number *N* can be evaluated in numerical simulations following a similar strategy as in the cited work on finite heat baths in optics and quantum molecular dynamics. The classical model of a bistable measurement process gives rise to sets of coupled Hamiltonian equations of motion that can be integrated using symplectic solvers.

In both cases, the immediate objective is to increase the mode number *N* as far as possible, in order to come close to an irreversible behavior, at least on time scales larger than all characteristic times of the object. The scenario sketched above for sufficiently high values of *N* is a plausible expectation, based on arguments involving analogies and extrapolating known results. It would relegate it to a similar category of practically incalculable quantum many-body phenomena such as, e.g., classical thermal fluctuations or Brownian motion, in short, reveal it as amplified quantum noise.

An unexpected but important consequence of this view is that it effectively merges the “first” and the “second” collapse of the wave packet into a single unitary process. In this way, it avoids the conceptually unsatisfactory detour from a pure initial state (a Schrödinger cat) to a mixture, after the first collapse, and back to a pure state (a definite measurement result) and in particular complies with entropy conservation throughout the entire measurement. In this way, also the second collapse becomes a process in continuous time, described by standard quantum mechanics, as it had already been ahieved for the first collapse.

Besides this central message, a unitary account of quantum measurement has various additional testable implications:The approach of the object state to one of the pointer states, as a final result of the measurement, will never be complete. In the limit N→∞, the discrepancy is expected to become arbitrarily small, but the postulate of pure states resulting from quantum measurement cannot be accomplished literally.Owing to the unavoidable entanglement between object and meter, the initial state of the meter does not only affect the final state of the object; the state of the object upon leaving the apparatus in turn also leaves a trace in the meter, which can then be probed by the following measurement. This implies the possibility of correlations between subsequent spin measurements, otherwise incompatible with their randomness, if their separation in time is extremely short.Spin measurements on systems prepared as Schrödinger cats with respect to the measured spin component are in the focus of this section. This notwithstanding, also “redundant” measurements, performed on systems that are prepared already with a definite polarization in the measured direction, are of interest in this context: The existence of a back-action of the meter on the object implies that even in the case of redundant measurements, albeit with very low probability, the measurement process could alter the spin polarization—trigger a spin flip—so that the result would not coincide with the state of the spin upon entering the apparatus.The approach outlined herein emphasizes the relevance of the meter state for the measurement outcome. Besides its initial state proper, this includes also invariant properties of the meter, such as its eigenenergy spectrum and the way it couples to the object. If, for example, the “meter” is represented by a microwave cavity, as is often the case in quantum optics, particular structures in the cavity spectrum will have an observable effect on the measurement results.In state-of-the-art laboratory experiments on quantum randomness [72], photons in counter-rotating polarization states replace the spins traditionally used as qbits in this context. It appears possible and tempting to work out the theory developed here so as to apply it to photon experiments.

Random spin measurements are almost invariably discussed in a special context where indeed they play a crucial rôle: Einstein–Podolsky–Rosen (EPR) experiments [88,89,90]. This issue has deliberately been avoided here, as it is charged with misleading connotations. In particular, in EPR experiments, quantum randomness is not only inextricably connected to nonlocality, it is even discussed as depending on it as on a necessary condition [72]. The present approach, however, is unrelated to this question, and it is not intended either to contribute in any sense to the long-standing debate around nonlocality and hidden-variable approaches. Yet, it cannot be denied that it has implications also for the interpretation of EPR experiments. Should it be the case that the meter has an impact on individual spin measurements, how then can spontaneous correlations arise between simultaneous measurements on spin pairs with a space-like separation? This issue should be relegated to future research as a particularly intriguing subject, to be addressed once the basic questions raised in this section have been settled.

## 5. Conclusions

The present report spans a wide arc, from minimalist models of chaos inspired by card shuffling, through pseudo-chaotic behavior in pixelated spaces, through the quantum death of classical chaos, through spin measurement. These diverse subjects do have a common denominator. They allow us to peek, from a macroscopic observation platform, into the details of information processing on the smallest scales, directing attention to a few essential aspects: fundamental limits of total information supply and storage density on these scales, “vertical” information currents interchanging entropy with large scales and “horizontal” exchange of entropy with adjacent degrees of freedom of the environment.

They are relevant in particular for an understanding of stochastic processes, collectively perceived as “randomness”, on the macroscopic level. The analysis presented here supports the view that they form exceptional points where information is not dumped into, but lifted up from small scales. While this idea may be little more than a helpful metaphor in the context of classical chaos, it suggests surprising consequences if applied to a seemingly unrelated field, quantum measurement. The randomness generated in quantum measurement can be seen in a similar spirit as resulting from an instability of the coupled object-meter system as it evolves towards alternative measurement results.

An interpretation and extrapolation of quantum chaos in this sense is but a single example of the fruitfulness of studying quantum phenomena in terms of information currents related to entanglement. This approach, originating in and inspired by the success of quantum information science applied to computing, is developing into an active research area of its own right, with applications in quantum optics, quantum many-body physics, and other areas waiting to be explored.

While entropy and information currents have proven invaluable tools to understand classical and quantum chaos, the discussion of randomness in quantum measurement reveals a significant shortcoming of quantum entropy as an analytical instrument: it is insensitive to the difference between ordered strings and random strings. Intuitively, a structural criterion for randomness should also be reflected in a suitable entropy measure for quantum processes, as it is indeed addressed on the classical level, notably in the context of algorithmic complexity [91,92,93,94]. 

## Figures and Tables

**Figure 1 entropy-21-00286-f001:**
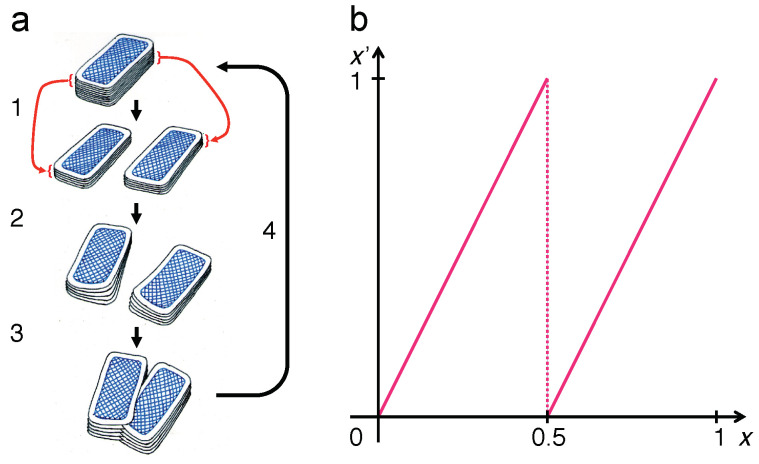
The Bernoulli map can be understood as modeling a popular card shuffling technique (**a**). It consists of three steps: (1) dividing the card deck into two halves of equal size, (2) fanning the two half decks out to twice the original thickness; and (3) intercalating one into the other as by the zipper method. (**b**) Replacing the discrete card position in the deck by a continuous spatial coordinate, it reduces to a map with a simple piecewise linear graph; cf. Equation (Equation 1).

**Figure 2 entropy-21-00286-f002:**
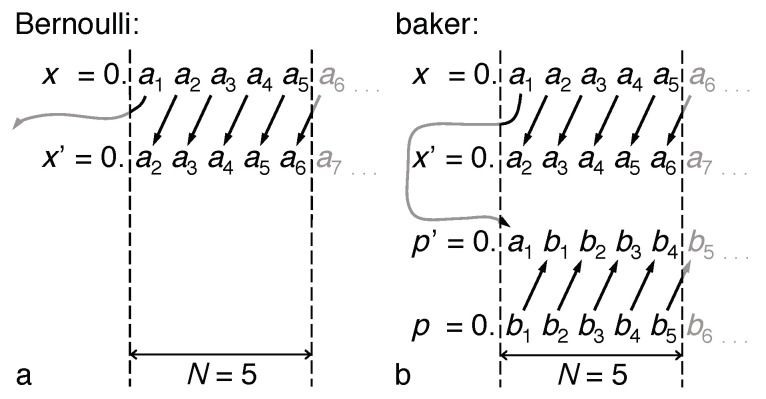
Representing the Bernoulli map (Equation (Equation 1)) in terms of its action on a symbol string; the position encoded as a binary sequence (see Equation (Equation 2) reveals that it corresponds to a rigid shift by one digit of the string towards the most significant digit (**a**). Encoding the baker map, Equation (Equation 5), in the same way, Equation (Equation 7), shows that the upward symbol shift in *x* is complemented by a downward shift in *p* (**b**). The loss of the most significant digit in the Bernoulli map or its transfer from position to momentum in the baker map is compensated by an equivalent gain or loss at the least significant digits, if a finite resolution is taken into account, here limiting the binary code to N=5 digits.

**Figure 3 entropy-21-00286-f003:**
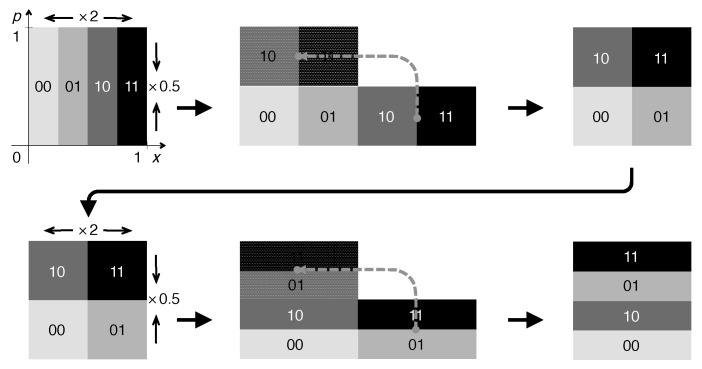
The baker map complements the Bernoulli map (Figure 1) by a coordinate *p*, canonically conjugate to the position *x*, so as to become consistent with symplectic phase-space geometry. Defining the map for *p* as the inverse of the Bernoulli map, a map of the unit square onto itself results (see Equation (Equation 5)) that is equivalent to a combination of stretching and folding steps. The figure shows two subsequent applications of the baker map and its effect on the binary code associated with a set of four phase-space cells.

**Figure 4 entropy-21-00286-f004:**
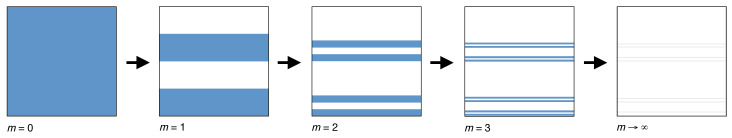
A dissipative version of the baker map is created by preceding each iteration of the map, as in Figure 3, with a contraction by a factor *a* in *p* (vertical axis), not compensated by a corresponding expansion in *x* (horizontal axis); see Equation (Equation 8). The figure illustrates this process for a homogeneous initial density distribution (m=0) and a contraction factor a=0.5 over the first three steps, m=1,2,3. Asymptotically for m→∞, under the alternation of contraction and splitting, the distribution condenses onto a strange attractor (rightmost panel) with a fractal dimension D=1.5.

**Figure 5 entropy-21-00286-f005:**
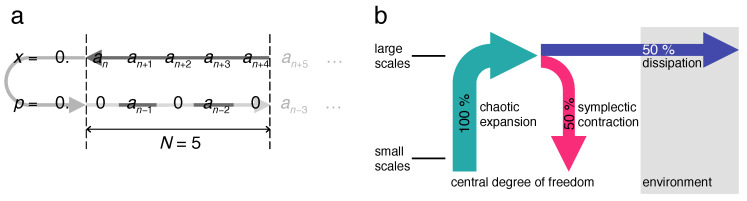
(**a**) In terms of binary strings that encode position *x* and momentum *p*, resp., including dissipative contraction by a factor a=0.5 in the baker map (see Figure 4), results in an additional digit zero fitted in between every two binary digits, transferred from the upward Bernoulli shift in *x* to the downward shift in *p*. (**b**) For bottom-up (green) and top-down (pink) information currents, this means that half of the microscopic information arriving at large scales by chaotic expansion is diverted by dissipation (blue) to the environment, thus returning to small scales in adjacent degrees of freedom.

**Figure 6 entropy-21-00286-f006:**
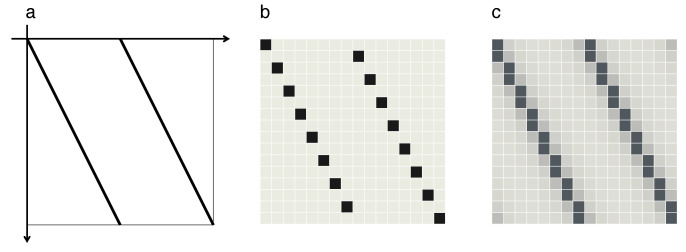
Three versions of the Bernoulli map exhibit a common underlying structure. The graph of the classical continuous map, Equation (Equation 1) (**a**) recurs in the structure of the matrix generating the discretized Bernoulli map (**b**), Equation (Equation 28), here for cell number J=16, and becomes visible, as well, as marked “ridges” in the unitary transformation generating (**c**) the quantum baker map, here depicted as the absolute value of the transformation matrix in the position representation, for a Hilbert space dimension $D_{H}=J=16$. The grey-level code in (**b**,**c**) ranges from light grey (zero) through black (one).

**Figure 7 entropy-21-00286-f007:**
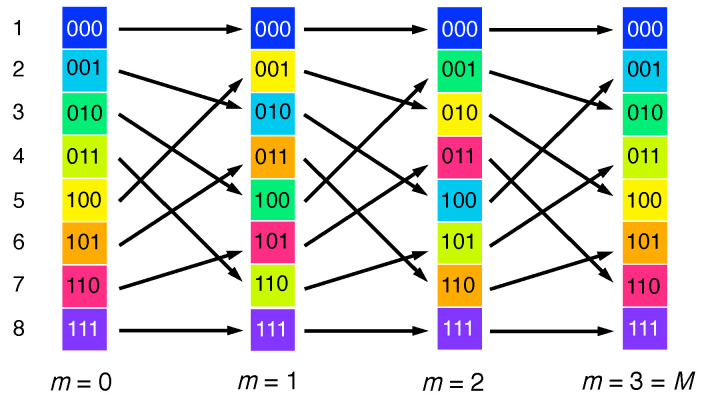
Accounting for the discreteness of the cards in the card-shuffling model (see Figure 1a) reduces the Bernoulli map to a discrete permutation matrix, Equation (Equation 28). The figure shows how it leads to a complete unshuffling of the cards after a finite number M=lb(J) of steps, here for M=3. Moreover, a binary coding of the cell index reveals that subsequent positions of a card are given by permutations of its three-digit binary code.

**Figure 8 entropy-21-00286-f008:**
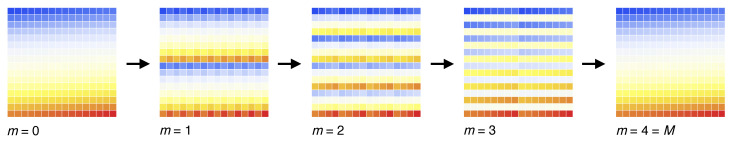
The recurrence in the discrete Bernoulli map (see Figure 7) occurs likewise in the discrete baker map, Equation (Equation 30). The figure shows how the simultaneous expansion in *x* (horizontal axis) and contraction in *p* (vertical axis) in the pixelated two-dimensional state space entail an exact reconstruction of the initial state, here after M=lb(16)=4 iterations of the map.

**Figure 9 entropy-21-00286-f009:**
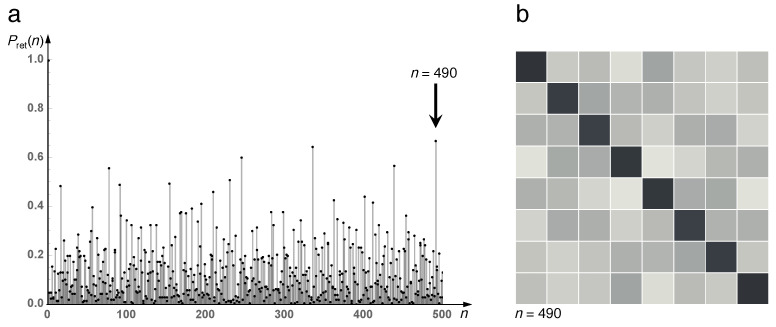
Recurrences in the quantum baker map are neither periodic, nor precise, as in the discretized classical version (see Figure 8), but occur as approximate revivals. They can be identified as marked peaks (**a**) of the return probability, Equation (Equation 37). For the strong peak at time n=490 (arrow in (**a**)), the transformation matrix in the position representation $B_{x,J}^{n}$ (**b**) (cf. Equation (Equation 36)), here with J=8, indeed comes close to a unit matrix. The grey-level code in (**b**) ranges from light grey (zero) through black (one).

**Figure 10 entropy-21-00286-f010:**
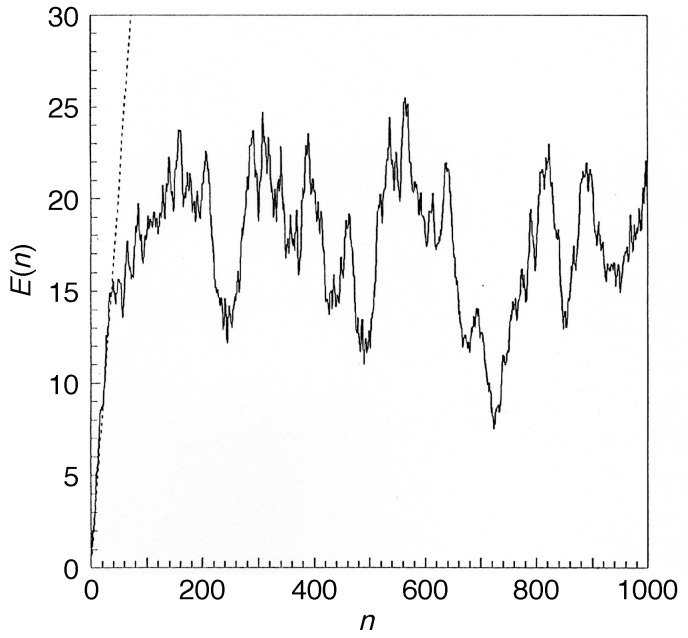
Suppression of deterministic angular momentum diffusion in the quantum kicked rotor. Time evolution of the mean kinetic energy, $E\left(n\right)=\langle p_{n}^{2}/2\rangle$, over the first 1000 time steps, for the classical kicked rotor, Equation (Equation 12) (dotted), and its quantized version, Equation (Equation 48) (solid line). The parameter values are K=10 and 2πℏ=0.15/G ($G:=(\sqrt{5}-1)/2$).

**Figure 11 entropy-21-00286-f011:**
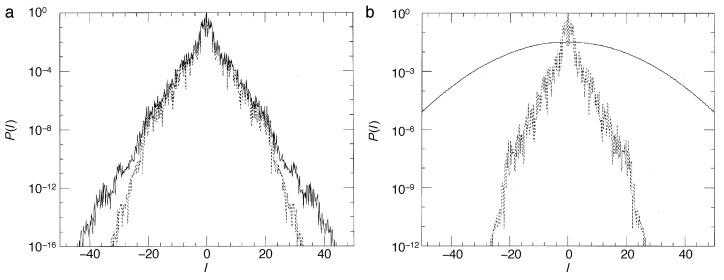
Dynamical localization is destroyed in the quantum kicked rotor with continuous measurements. Probability distribution P(l) of the angular momentum *l* (semilogarithmic plot), after the first 512 time steps, for the measured dynamics of the quantum kicked rotor, Equation (Equation 48) (solid lines), compared to the unmeasured dynamics of the same system, Equation (Equation 48) (dashed), for (**a**) weak vs. (**b**) strong effective coupling. A continuous measurement of the full action distribution was assumed. The parameter values are K=5, 2πℏ=0.1/G ($G:=(\sqrt{5}-1)/2$), and $\nu =10^{-4}$ (**a**), ν=0.5 (**b**). Reproduced from data underlying [59].

**Figure 12 entropy-21-00286-f012:**
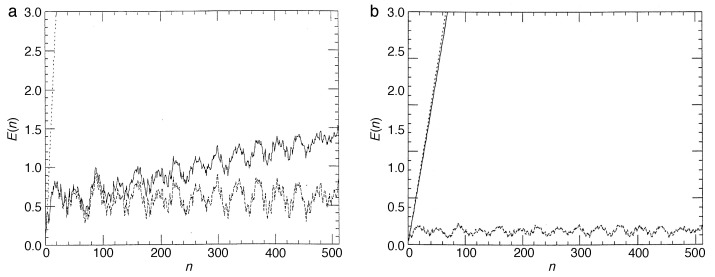
Deterministic angular momentum diffusion is revived in the quantum kicked rotor with continuous measurements. Time evolution of the mean kinetic energy, $E\left(n\right)=\langle p_{n}^{2}/2\rangle$, over the first 512 time steps for the measured dynamics of the quantum kicked rotor, Equation (Equation 48) (solid line), the stochastic classical map, Equations (Equation 82) and (Equation 83) (dotted line), and the unobserved dynamics of the quantum kicked rotor, Equation (Equation 48) (dashed line), for (**a**) weak vs. (**b**) strong effective coupling. A continuous measurement of the full action distribution was assumed. The parameter values are K=5, 2πℏ=0.1/G ($G:=(\sqrt{5}-1)/2$), and $\nu =10^{-3}$ (**a**), ν=0.5 (**b**).

**Figure 13 entropy-21-00286-f013:**
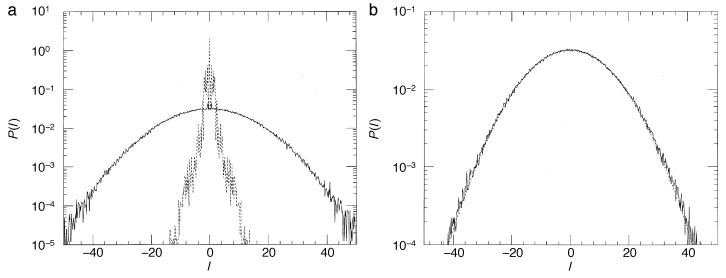
Same as Figure 11, but comparing the measured dynamics of the quantum kicked rotor, Equation (Equation 48) (dashed lines), to the stochastic classical map, Equations (Equation 82) and (Equation 83) (solid lines), for (**a**) weak vs. (**b**) strong effective coupling. A continuous measurement of the full action distribution was assumed. The parameter values are K=10, 2πℏ=0.1/G ($G:=(\sqrt{5}-1)/2$), and $\nu =10^{-4}$ (**a**), ν=0.5 (**b**). Reproduced from data underlying [59].

**Figure 14 entropy-21-00286-f014:**
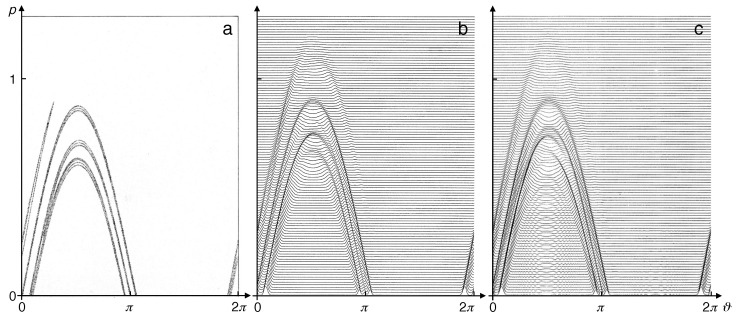
Classical and quantum stationary-state distributions of the dissipative standard map for n≫1/λ. (**a**) Support of the strange attractor of the classical map (Equation 23) in (p,θ) phase-space. (**b**) Stationary state of the semiclassical stochastic map (Equation 50), plotted at discrete angular momentum values $p_{l}=\hbar l$, as in panel (**c**). (**c**) Long-time limit of the density operator for the master equation (Equation 49), represented as the corresponding Wigner function, which has support along the quantized angular momentum values lℏ, l∈Z. The parameter values are n=10, K=5, λ=0.3, and 2πℏ=0.02 (**b**,**c**). Only the upper (positive-momentum, p≥0) part of phase space is shown, the lower (p≤0) part is related to it by parity, p→−p, θ→−θ. Reproduced from data underlying [71].

**Figure 15 entropy-21-00286-f015:**
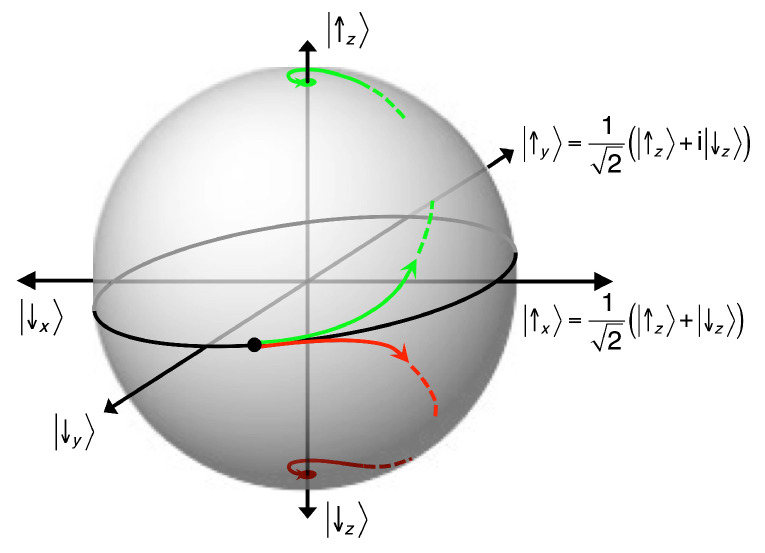
Spin measurement on the Bloch sphere. The quantum dynamics of spin measurements is dominated by two “pointer states”, eigenstates of the measured operator ${\hat{\sigma }}_{z}$, i.e., ${|\;↑}_{z}\rangle$ and ${|\;↓}_{z}\rangle$, represented on the Bloch sphere as North (green dot) and South pole (red dot). Owing to the quantum Zeno effect, they attract nearby states of the measured system. At the same time, the short-time evolution of the measured spin for a meter comprising only a single boson mode, Equation (Equation 72), suggests that a state initiated on the equator of the Bloch sphere (black dot), besides rotating around the equator, will tend towards one of the poles, depending on the initial state of the meter boson mode.

**Figure 16 entropy-21-00286-f016:**
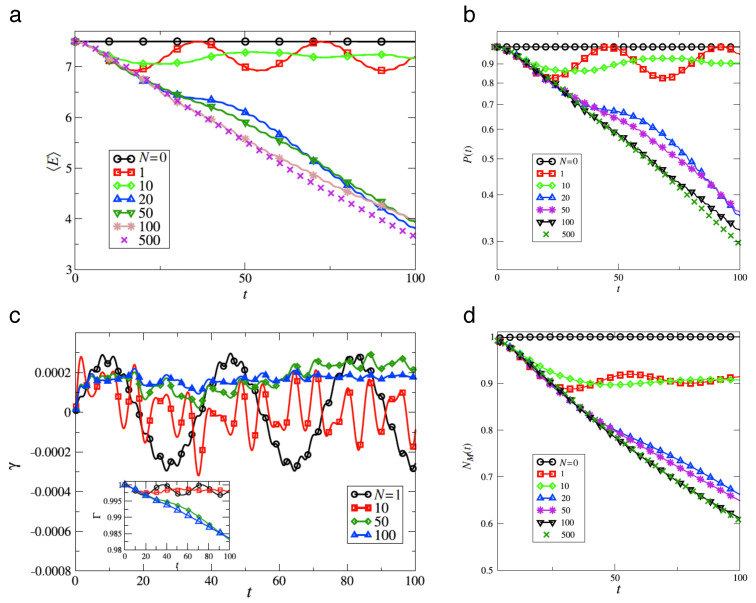
Decoherence-like behavior can be simulated by coupling a harmonic oscillator to a reservoir comprising only a finite number *N* of boson modes (harmonic oscillators, as well). The figure shows the time evolution of four diagnostics of decoherence for different values of *N*, ranging from N=0 (isolated central system) through 10, 20, 50, 100, through 500 (see legend). (**a**) Total energy in the central system, showing a crossover from exponential to power-law decay for N≥10. (**b**) Purity $P\left(t\right)=\mathrm{Tr}[{\left(\rho_{O}\left(t\right)\right)}^{2}]$. (**c**) Instantaneous dissipation rate, i.e., ratio of effective friction force to time-dependent velocity (inset: total energy as in panel (**a**)), for N=1, 10, 50, 100. (**d**) Degree of memory, measured as the non-Markovianity $N_{M}\left(t\right)=\frac{1}{t}\int_{0}^{t}dt^{\prime }\;\left|P\left(t^{\prime }\right)\right|$, P(t) denoting the purity as depicted in panel (**b**). Reproduced from [81] with kind permission.

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
