# Peer review of "Quantum Chaos and Quantum Randomness—Paradigms of Entropy Production on the Smallest Scales"

_entropy, 2019, doi:10.3390/e21030286_

Round 1

Reviewer 1 Report

This ambitious manuscript tells a story of chaos, randomness and entropy in classical and quantum systems, and posits that the backflow of entropy form the measurement process could be the source of the indeterminancy of quantum mechanics. Without passing judgement on that last speculation (which the author couches cautiously), I enthusiastically support publication.

I think the Introduction and Conclusion are exceedingly clear, and Sections 2 and 4 are efficient at making technical arguments plausible. I felt Section 3 was less successful, partially due to notational glitches described below, but also for two other reasons: 1) the through-line of the argument was less clearly signposted and structured than the other sections; 2) more intuition/knowledge was assumed (particularly about localization) than in the other sections. I would recommend the author re-reading this section with a fresh eye, and seeing whether small changes might make the argument more cogent for more readers. In particular, I found between eq. (43) and (44) quite dense.

But again, I learned a great deal from the article and highly recommend it for the journal Entropy. Below are a few more itemized comments, mostly typographical

* On the first page, on line 9 and 24 the closing parentheses are replaced by quotes.

* At line 388, there is a missing parenthesis in the exponent of $B_l^{(p)}$.

* In equation (41), the definition of $\hat{U}_{\mathrm{rot}}$ includes the notation for the angular momentum operator $\hat{l}$ that isn't actually introduced until the text above eq. (48). At this point, $\hat{p}$ is still being used for angular momentum, I believe. The notation and physical interpretation for the central value $l_c(\epsilon)$ at the end of the page was also introduced somewhat ad hoc. The following paragraphs about localization were not obvious at all and I wish there was someway the author could help me understand this better.

*At line 417, there is missing space between "constant mean".

*Line 546 isn't a complete sentence.

*I would have appreciated a summary of Section 3 recounting the structure of the argument and highlighting key results. Perhaps the paragraphs starting around 617 are meant to accomplish this, but I missed it.

*Near the end of the caption for Fig. 14 there is an equation number (pleq0) not correctly coded.

*At line 689 there is a miscoded $\psi$.

*At line 753, I somewhat dispute introducing the purity of the reduced density matrix with the phrase "more specifically".

Author Response

I am grateful to both referees for their willingness to review this exceptionally long manuscript
under a tight schedule just after Christmas vacations.

This paper is a daring endeavour. Encouragement by a knowledgeable reviewer is
appreciated indeed.

Following the referee’s report, I have above all rewritten long passages of section 3,
amending confusing formulations, complementing incomplete arguments, and making
my reasoning more explicit where necessary. In particular, the paragraphs on dynamical
localization have been reformulated, and intermediate summaries have been added
to anchor details in the main thread of my reasoning. At the same time, I tried to keep
the unavoidable increase in length of this section within narrow limits. I hope that
as a result, this section has become more readable and comprehensible.

All the typos and syntax errors pointed out by this referee have been eliminated,
together with numerous others I became aware of during the revision.

Changes in response to referee 1:

Large passages in section 3 have been reformulated or added, in particular

p. 15 l. 410ff, p. 13 l. 343/344, p. 15 l. 410ff, p. 16 l. 436ff and l. 447ff, p. 17 l. 481ff and l. 490ff,
p. 22 l. 624ff and l. 629ff.

All typos, syntax errers etc. pointed out by the referee have been eliminated, in particular

p. 1 l. 9 and l. 24, p. 17 l. 394, p.15 eq.(37) and l. 434, pp. 15,16,17 major changes /see above),
p. 22 caption fig.14, p. 24 l. 69, p.34 l. 1015 (deferred from l. 546, original version)

Reviewer 2 Report

attached report

Author Response

I am grateful to both referees for their willingness to review this exceptionally long manuscript
under a tight schedule just after Christmas vacations.

To introduce my response to Referee 2, I would like to resume briefly the rationale
of this paper:

It is intended as an invitation to interpret quantum chaos as a phenomenon revealing,
more than other subjects within quantum mechanics, the particular way quantum systems
process entropy. It thus takes up the theme set by the seminal 1981 article by Robert Shaw
on chaotic behaviour and information flow (my Ref. [2]) and extends it to chaos in quantum
systems. This explains the review-like character of sections 2 and 3. Section 4 elaborates on one
of the main conclusions of the foregoing sections and applies it to a field not directly
related to quantum chaos, to quantum measurement, to demonstrate the explanatory power
of the information theoretic conclusions. It may appear speculative, but in fact proposes
a specific model to substantiate these argumemts, together with analytic results that are available
right now, before they can be complemented with numerical simulations. In this strategy, the paper
follows another relevant reference, the article by Raimond, Brune, and Haroche (my Ref. [67]),
pointing out an interpretation of decoherence as an, in principle, reversible process.

Trying to extract a central message from this report, I had the impression that the referee
had preferred to see a paper questioning the concept of quantum chaos from the outset.
In this point, I have difficulties to comply with her or his expectation: I have been invited to
contribute a paper to a special issue on “Quantum Chaos and Complexity” of a journal
entitled “Entropy”. With my contribution, I intended to substantiate the tight relationship
between entropy and quantum chaos. Denying the very existence of the concept
that forms half of the title of that special issue would not have appeared to me as
an adequate response to that kind invitation.

The referee expresses his surprise that the argument asserting that “quantum mechanics,
being at base a linear theory, does not have real ‘chaos.’ “ has not been referred to
in the manuscript. The reason is not that I had not been aware of it, but that I am convinced
that it is not correct. The counterargument is similarly simple, invoking a reductio ad absurdum:

If classical equations of motion are reformulated in a mathematical language similar to that
of Schrödinger’s equation—in terms of a density defined on phase space—they take the form
of Liouville’s equation. It is equivalent to Hamilton’s equations and predicts the same physics,
but is perfectly linear in the probability density, as is Schrödinger’s equation in the wavefunction.
If this linearity was to prevent chaotic behaviour, it could not exists in classical mechanics,
either.

This refutation is actually quite well-known in the community, so that I did not consider
including it in my paper. However, it shows that another general argument is wanted that
would summarize the evident absence of chaos in closed quantum systems, explained by
specific technical arguments like dynamical localization, in each individual instance,
in a general reasoning. I am convinced and elucidate it in my paper, that basic constraints
on the content and representation of information in quantum systems provide such
an explanation.

I appreciate the remarks and suggestions as to the literature cited in the paper. However,
in view of its information theoretic focus, I do not find it adequate to provide with it a broad
bibliography on quantum chaos in general. In particular, the authors recommended
by the referee (Dominique Delande, Peter M. Koch, Bala Sundaram, Dieter Wintgen) have
published mainly on applications of quantum chaos in atomic and solid-state physics,
areas too far from the subject of the paper to even find a suitable anchor point to cite them.
Instead, I include references to a few selected monographs on quantum chaos which
already contain comprehensive bibliographies.

Reacting on this referee’s criticism concerning general style and attitude of the manuscript,
I have deleted too fancy metaphors or too far-fetched speculation throughout section 4.

Responding to the referee’s specific comments 1 to 6,

1
Sections 2 and 3 have review character, but do contain some original material,
in particular on the discrete Bernoulii and Baker maps, as well as fresh data on
the quantum baker map. In order to present it, the corresponding continuous
and classical sysrems, resp., must have been introduced before.

2
The fact that quantum measurement not only requires a transfer of information
from the object to the apparatus (that’s its purpose), but entails a transfer als
in the opposite direction is recognized in numerous publicatipons on the subject,
some of them cited in my paper. The key to substantiate this statement lies in
the fact that object and meter get entangled during the measurement, so that
they share their entropy: The entropy of the total system cannot be partitioned
uniquely between its two components. When they get disentagled again after the
measurement, there is no reason why the entropy should be segmented along
exactly the same lines as it had been before the measurement. The meter will
remain with information on the object, which in turn will leave with some information
on the initial state of the meter.

Calling this process a “flow” or an “infiltration” may not hit exactly the meaning
of the abstract concept of entanglement, but I think they are justfied for
the sake of a clear graphical language.

3
The passage referred to by the referee has been reformulated to clarify the
use of the terms “top-down” and “bottom-up” information current.

4
The “second collapse”, formalized in Eq. (67) (original version), is usually
taken for granted as it appears as a mere classical random process. However,
in the context of the paper, it is highly relevant since it amounts to a step from a
mixed state resulting from the entanglement of the object with the meter
(non-zero von-Neumann entropy) back to a pure state (vanishing entropy).
In this sense, no entropy is gained nor lost with respect to the initial state,
but there is a loss compared to the result of the first collapse. At the same time,
the second collapse, in particular in in spin measurement ,is the source of
“quantum randomness”, a valuable resource of entropy that even forms
a commodity for the IT business. In this sense, the step described by Eq. (67)
*creates* entropy instead of reducing it, as implied by the Copenhagen
interpretation. Section 4 attempts to resolve this striking inconsistency,
extending the picture to the combined system, object + meter, and the entropy
conservation valid for this comprehensive system.

I have made this argument more explicit in two text passages added to the
general discussion of quantum measurement.

5
I completely agree with the referee in this point. However, there is no doubt either
that the intermediate state resulting from the first collapse (two eigenvalues 1/2 of
the 2x2 density matrix) corresponds to an entropy of one bit. Concerning the second
collapse, different interpretations exist, see above.

6
Equation (91) (original version) refers to the case with Ohmic friction, as it results
in the limit N –> to \infty of the number of heat bath oscillators. This limit is a known
result of microscopic models for friction and Brownian motion on the classical level
(see, e.g., the work by Uhlenbeck and Gousmit). The two potential terms containing
parameters a and b are not putnin by hand but are just the negative gradient of the
potential specified with the potential.

Changes in response to referee 2:

Text passages in sections 2 to 4 have been rephrased, in particular

p. 3 l. 116ff, p. 12 l. 336/337, p. 17 l. 490ff, p. 23 l. 666ff, p. 24, l. 678ff

In sections 4 and 5, statements of speculative character, overly fancy metaphors, etc. have been
deleted, in particular

p. 27 l. 797, p. 30 l. 863,865,872,891,904, p. 31 l. 907,921,925.

Round 2

Reviewer 2 Report

The author has inserted (in blue) sentences and paragraphs but, as in his response letter to my report, we seem still far apart on both specifics and general subject. I do not see any further purpose to continuing the exchange but do wish to note that a main point of his rebuttal, that classical physics can be rendered in Liouville form does not address that there is nevertheless a fundamental distinction between classical and quantum physics/mechanics. This was my main objection pertaining to use of "quantum chaos." That Planck's constant is finite and nonzero (even if small) makes a fundamental difference, that the superposition principle applies at the fundamental level and thus there is no nonlinearity, that whether in turbulence or the classic Lorentz study of chaos some essential nonlinearity is necessary, have not been refuted. Even in his response, in directing readers on line 337 to refs 34-36, note the very title of the last one of those: "quantum signatures of chaos." This is what I had pointed to, that a consensus had developed that one can look for how systems whose classical counterparts are chaotic manifest in the corresponding quantal system, words such as chaology or signatures in distribution statistics being used, but the pairing "quantum chaos" is self-contradictory. 

       I note now from his response that this paper seems to have been solicited for an issue on chaos, and I certainly do not see my role as referee to stand in the way. Rather, I wish to point out the problematic aspects and the misleading one to readers that quantum chaos exists, and leave it to author and editor on playing a role in further propagating such an idea.